# SALIENCY-GUIDED HIDDEN ASSOCIATIVE REPLAY FOR CONTINUAL LEARNING

## ABSTRACT

Continual Learning (CL) is a burgeoning domain in next-generation AI, focusing on training neural networks over a sequence of tasks akin to human learning. While CL provides an edge over traditional supervised learning, its central challenge remains to counteract *catastrophic forgetting* and ensure the retention of prior tasks during subsequent learning. Amongst various strategies to tackle this, replay-based methods have emerged as preeminent, echoing biological memory mechanisms. However, these methods are memory-intensive, often preserving entire data samples—an approach inconsistent with humans' selective memory retention of salient experiences. While some recent works have explored the storage of only significant portions of data in episodic memory, the inherent nature of partial data necessitates innovative retrieval mechanisms. Current solutions, like inpainting, approximate full data reconstruction from partial cues, a method that diverges from genuine human memory processes. Addressing these nuances, this paper presents the **S**aliency-Guided **H**idden **A**ssociative **R**eplay for **C**ontinual Learning (**SHARC**). This novel framework synergizes associative memory with replay-based strategies. SHARC primarily archives salient data segments via sparse memory encoding. Importantly, by harnessing associative memory paradigms, it introduces a content-focused memory retrieval mechanism, promising swift and near-perfect recall, bringing CL a step closer to authentic human memory processes. Extensive experimental results demonstrate the effectiveness of our proposed method for various continual learning tasks.

## 1 INTRODUCTION

Continual learning (CL) represents a vital advancement for next-generation AI, allowing neural networks to sequentially learn tasks like humans do Parisi et al. (2019). While traditional supervised learning is well-established, CL remains in its nascent stages. The main challenge is to prevent *Catastrophic Forgetting* McCloskey & Cohen (1989) as agents acquire new tasks, ensuring they retain earlier knowledge. In essence, CL strives to balance updating the model with retention across a series of tasks. In order to address this problem, researchers have put forward several strategies. *Replay-based methods* Rebuffi et al. (2017); Aljundi et al. (2019); Arani et al. (2022), which utilizes a small memory to store previous data and reuse them when learning new tasks, have emerged as a particularly effective solution, offering superior performance and drawing inspiration from biological systems Robins (1995). However, a potential bottleneck of this approach is its memory-intensive nature, as entire data samples are conserved. This mechanism contrasts starkly with the human brain's approach to memory retention. Humans typically do not remember every detail but tend to recall fragments or the most salient features of experiences Rolls (2013). The vast storage requirements of replay-based methods and their divergence from natural memory processes necessitate exploration into more efficient and human-like strategies for continual learning.

While there are pioneering works Saha & Roy (2023); Bai et al. (2023) in replay-based CL that have begun to explore the idea of storing only the salient or partial aspects of data into episodic memory, challenges arise due to the inherent nature of partial data. Since these fragments are not directly usable as model input, an effective retrieval technique becomes indispensable. A straightforward solution is inpainting Elharrouss et al. (2020), which, through rule-based or generative models, attempts to recreate the full data from the available partial cue. This methodology, however, essentially approximates the entirety of the data by generating similar samples from a given distribution and may suffer from inaccurate retrieval under large noise or corruption (Col 3 in Figure 1). On the contrary,

| *Original Images* | *Query Images* | *Generative Model* | *Associative Memory* |
|---|---|---|---|

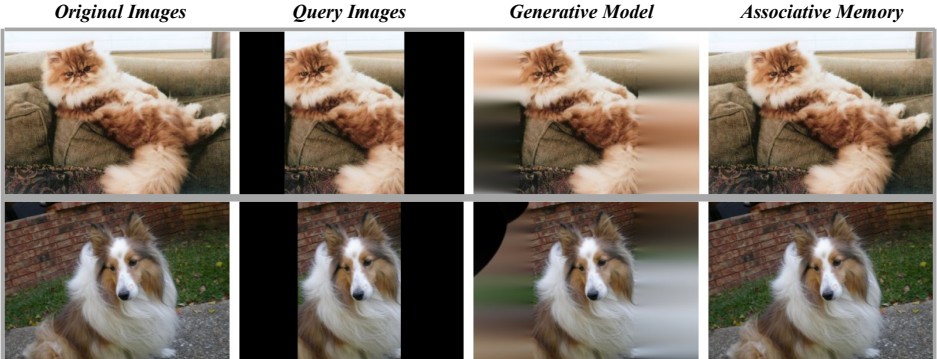

Figure 1: Content-based (associative memory) *v.s.* generative model retrieval. Pixels from non-salient areas are masked in query images. For a fair comparison, we train an autoencoder-based inpainting model Peng et al. (2021) and a Hopfield Network Ramsauer et al. (2020) with similar number of parameters. Associative memory achieves almost perfect recall even under large noise or corruption.

the human brain, especially the hippocampus, employs associative recall for *content-based* memory retrieval Hopfield (1982); Ramsauer et al. (2020), achieving a remarkable recall accuracy close to perfection (Col 4 in Figure 1). As such, for systems aiming to emulate human-like continual learning, there is an evident inspiration to design techniques that mirror the associative and content-based retrieval processes inherent in human cognition.

To address the aforementioned challenges, this paper introduces the **S**aliency-Guided **H**idden **A**ssociative **R**eplay for **C**ontinual Learning (**SHARC**), marking the inception of a Continual Learning framework that seamlessly integrates associative memory into replay-based techniques. As depicted in Figure 1, SHARC distinguishes itself from existing replay-based CL methodologies in two pivotal aspects: First, rather than archiving complete samples within episodic memory, SHARC conserves only the most salient segments through sparse memory encoding. More crucially, drawing inspiration from the principles of associative memory, we have crafted a content-centric memory retrieval module that boasts swift and impeccable recall capabilities.

Our contribution includes, 1). We develop a novel neural-inspired replay-based continual learning framework to handle catastrophic forgetting. 2). We propose to leverage associative memory for efficient memory storage and recovery. 3). We demonstrate our model's efficacy and superiority with extensive experiments.

## 2 RELATED WORK

**Continual Learning (CL).** Catastrophic forgetting is a long-standing problem Robins (1995) in continual learning which has been recently tackled in a variety of visual tasks such as image classification Kirkpatrick et al. (2017); Rebuffi et al. (2017), object detection Zhou et al. (2020), etc.

Existing techniques in CL can be divided into three main categories Parisi et al. (2019): 1) *regularization-based approaches*, 2) *dynamic architectures* and 3) *replay-based approaches*. Regularization-based approaches alleviate catastrophic forgetting by either adding a regularization term to the objective function Kirkpatrick et al. (2017) or knowledge distillation over previous tasks Li & Hoiem (2017). Dynamic architecture approaches adaptively accommodate the network architecture (e.g., adding more neurons or layers) in response to new information during training. Dynamic architectures can be explicit if new network branches are grown, or implicit, if some network parameters are only available for certain tasks. Replay-based approaches alleviate the forgetting of deep neural networks by replaying stored samples from the previous history when learning new ones and have been shown to be the most effective method for mitigating catastrophic forgetting.

**Replay-based CL.** Replay-based methods mainly include three directions: rehearsal methods, constrained optimization, and pseudo rehearsal. *Rehearsal methods* directly retrieve previous samples from a limited size memory together with new samples for training Chaudhry et al. (2019); Hayes et al. (2020); Arani et al. (2022). While simple in nature, this approach is prone to overfitting the old samples from the memory. As an alternative, *constrained optimization* methods formulate

backward/forward transfer as constraints in the objective function. GEM Lopez-Paz & Ranzato (2017) constrains new task updates to not interfere with previous tasks by projecting the estimated gradient on the feasible region outlined by previous task gradients through first-order Taylor series approximation. A-GEM Chaudhry et al. (2018) further extended GEM and made the constraint computationally more efficient. Finally, *pseudo-rehearsal* methods typically utilize generative models such as GAN Goodfellow et al. (2020) or VAE Pu et al. (2016) to generate previous samples from random inputs and have shown the ability to generate high-quality images recently Robins (1995). Readers may refer to Parisi et al. (2019) for a more comprehensive survey on continual learning.

**Associative Memory (AM).** In general, the attractor-based mechanism Amit & Amit (1989) is typically used for the implementation of AMs, which are models that store and recall patterns. Pattern recall (associative recall) is a process whereby an associative memory, upon receiving a potentially corrupted memory query, retrieves the associated value from memory. One of the earliest and probably the most well-known associative memory are Hopfield Networks Hopfield (1982). Hopfield networks are a class of recurrent artificial neural networks that have gained prominence for their ability to model associative memory and pattern recognition. The modern Hopfield network refers to an updated version of the original Hopfield network Ramsauer et al. (2020); Krotov & Hopfield (2020). The modern Hopfield network incorporates enhancements and modifications to improve its performance and overcome some limitations of the original model. More recently, predictive coding networks Huang & Rao (2011) have provided a new perspective for the design of AM, and such works Salvatori et al. (2021); Yoo & Wood (2022) have shown strong performance on recall tasks.

## 3 PROBLEM FORMULATION

We consider supervised continual learning in this paper. Following the learning protocol in Chaudhry et al. (2018), we consider a training set $\mathcal{D} = \{\mathcal{D}_1, \mathcal{D}_2, \cdots, \mathcal{D}_T\}$ consisting of $T$ tasks, where $\mathcal{D}_t = \{(\mathbf{x}_i^{(t)}, \mathbf{y}_i^{(t)})\}_{i=1}^{n_t}$ contains $n_t$ input-target pairs $(\mathbf{x}_i^{(t)}, \mathbf{y}_i^{(t)}) \in \mathcal{X} \times \mathcal{Y}$. While each learning task arrives sequentially, we make the assumption of *locally i.i.d*, i.e., $\forall\, t, (\mathbf{x}_i^{(t)}, \mathbf{y}_i^{(t)}) \overset{iid}{\sim} P_t$, where $P_t$ denotes the data distribution for task $t$ and $i.i.d$ for *independent and identically distributed*. Given such a stream of tasks, the goal is to train a learning agent $f_{\boldsymbol{\theta}} : \mathcal{X} \to \mathcal{Y}$, parameterized by $\boldsymbol{\theta}$, which can be queried *at any time* to predict the target $\mathbf{y}$ given associated unseen input $\mathbf{x}$ and task id $t$. Moreover, we require that such a learning agent can only store a small amount of seen samples in an episodic memory $\mathcal{M}$ with a fixed budget. Given predictor $f_{\boldsymbol{\theta}}$, the loss on the episodic memory of task $k$ is defined as

$$\ell(f_{\boldsymbol{\theta}}, \mathcal{M}_k) \coloneqq |\mathcal{M}_k|^{-1} \sum\nolimits_{(\mathbf{x}_i, k, \mathbf{y}_i)} \phi(f_{\boldsymbol{\theta}}(\mathbf{x_i}, k), \mathbf{y}_i), \ \forall\, k < t, \tag{1}$$

where $\phi$ can be *e.g.* cross-entropy or MSE. In general, a large body of *replay-based* continual learning methods seeks to optimize for the following loss function at $t$-th task

$$\min\nolimits_{\boldsymbol{\theta}} \mathcal{L}_{CL}(\boldsymbol{\theta}), \ \text{where } \mathcal{L}_{CL}(\boldsymbol{\theta}) = \sum\nolimits_{(\mathbf{x}, t, \mathbf{y})} \ell(f_{\boldsymbol{\theta}}(\mathbf{x}, t), \mathbf{y}) + \sum\nolimits_{k < t} \ell(f_{\boldsymbol{\theta}}, \mathcal{M}_k), \tag{2}$$

which is an aggregation of the losses on the current task and replay data. After the training of task $t$, a subset of training samples will be stored in the episodic memory, i.e., $\mathcal{M} = \mathcal{M} \cup \{(\mathbf{x}_i^{(t)}, \mathbf{y}_i^{(t)})\}_{i=1}^{m_t}$, where $m_t$ is the memory buffer size for the current task.

## 4 PROPOSED METHOD

In this section, we introduce our proposed Saliency-Guided Hidden Associative Replay for Continual Learning. We innovatively utilize saliency methods to select the most important channels of feature maps for each image and only store those channels in the episodic memory, thus achieving controllable and better memory efficiency. During the training phase, we leverage pattern association techniques for memory completion, where each partially stored image will be restored by a brain-inspired associative memory. An overview of our framework is shown in Figure 2.

### 4.1 SALIENCY-GUIDED MEMORY ENCODING WITH STRUCTURED SPARSITY

In this section, we discuss the memory encoding process of our method. According to the *hippocampal indexing theory* Teyler & Rudy (2007), there are two major characteristics of how the human brain encodes its memories. First, the encoded representations in the human hippocampus are highly *sparse*, meaning that only a small subset of neurons in the hippocampus is activated for each specific

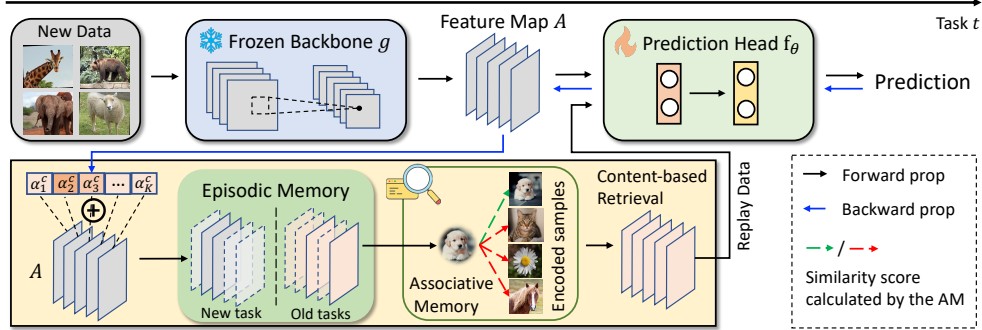

Figure 2: Overview of our proposed SHARC framework (best viewed in color). When new data comes, a pre-trained backbone is used to extract feature maps. Then, the saliency score of the saliency map is calculated via backpropagation and we drop channels with lower saliency, thus achieving structured sparsity and memory efficiency. During memory replay, previous feature maps are retrieved via an associative memory, which essentially picks the top-1 stored feature maps based on the similarity with the query feature map.

memory. Second, the stored representations for replay are not exact reproductions (e.g., raw pixels) Ji & Wilson (2007); instead, its visual inputs originate higher in the visual processing hierarchy rather than from the primary visual cortex or the retina Insausti et al. (2017). Motivated by this, our goal is to propose a computational model for memory encoding that satisfies both characteristics.

Different from many earlier works that store raw images for replay Lopez-Paz & Ranzato (2017); Ebrahimi et al. (2021); Arani et al. (2022), we consider first encoding raw data into high-level representations and store them. Formally, our model $\mathbf{y} = f_\theta(g(\mathbf{x}))$ is composed of a pre-trained backbone $g$ and a trainable prediction head $f_\theta$. In this work, the pre-trained backbone is general and can be instantiated as various vision models such as VGG, ResNet, and ViT.

The output of $g(\mathbf{x})$ is a tensor $A \in \mathbb{R}^{H \times W \times K}$, where $H, W$ are the dimension of the feature map and $K$ is the number of channels. To achieve sparse representation, we consider saliency-based methods Selvaraju et al. (2017) that measure the importance of the neurons by their first-order gradients. Specifically, the saliency approach computes the gradient of the score for class c, i.e., $f_\theta^{(c)}(A)$, with respect to feature map activations $A$, followed by a global-average-pooling over the width and height dimension to obtain the neuron importance weights $\alpha_k^c$:

$$\forall k, \quad \alpha_k^c = \frac{1}{H}\sum_{i=1}^{H}\frac{1}{W}\sum_{j=1}^{W}\partial f_\theta^{(c)}(A)/\partial A_{ij}^k. \tag{3}$$

Intuitively speaking, feature maps with higher magnitudes in the saliency are more likely to be involved in the region of the target class object while those with lower magnitudes are more likely to be the non-target objects or background regions. In addition, existing work Sun et al. (2015) has proved that hidden representation learned by convolutional neural networks is highly sparse in the hidden space. In this work, we consider the saliency score as a measure and mask out feature maps with lower saliency scores. Formally, denote $\boldsymbol{\alpha}^c = [\alpha_1^c, \alpha_2^c, \cdots, \alpha_K^c]$, the masked feature map $A'$ is

$$A' = \text{TR}_{H,W,K}\big(\mathbb{1}\{|\boldsymbol{\alpha}^c| > Q_\mu\} \otimes \mathbf{J}_{H,W}\big) \odot A, \tag{4}$$

where $\mathbb{1}\{\cdot\}$ is the indicator function, $Q_\mu$ is the threshold for masking out the bottom $\mu$ quantile of channels. $\mathbf{J}$ denotes the all-one matrix and $\text{TR}(\cdot)$ denotes tensor reshaping operation.

One advantage of our design is that the channel-wise operations result in **structured sparsity**, which is hardware-friendly and can lead to memory cost reduction instantly without further system-level efforts. To see this, we discard those channels that have a saliency score lower than the threshold, and the rest feature maps have a regular tensor shape. We only need to keep track of the channel index which is only a 1d vector and cheap to store.

## 4.2 ASSOCIATIVE MEMORY RETRIEVAL FOR REPLAY

In this section, we discuss the memory retrieval process of our method. Associative memory plays an important role in human intelligence and its mechanisms have been linked to attention in machine learning Ramsauer et al. (2020). Recently, the machine learning community's interest in associative

---

**Algorithm 1** SHARC Training

---

**Require:** Continual learning classifier $f_\theta$, associative memory $\mathcal{A}(\cdot, \boldsymbol{\omega})$, training continuum $\mathcal{D}^{train}$, dropping threshold $\mu$, optimizer OPT, forgetting frequency $R$, total number of tasks $T$.
1:  $\mathcal{M}_t \leftarrow \{\}, \forall\, t = 1, 2, \cdots, T$          ▷ Initialize episodic memory
2:  **for** $t = 1$ **to** $T$ **do**
3:      $\tilde{\mathcal{M}}_k \leftarrow \mathrm{OPT}_\mathbf{x}(\mathbf{x}, \mathcal{M}_{k<t}, \boldsymbol{\omega}),\ \forall\, k < t$ as Eq. 2      ▷ Associative memory *read*
4:      **for** $\mathcal{B}_t \sim \mathcal{D}_t^{train}$ **do**
5:         $\theta \leftarrow \mathrm{OPT}_\theta(\theta, \mathcal{B}_t, \tilde{\mathcal{M}}_{k<t})$ as Eq. 2      ▷ Train the classifier
6:         **for** $(x, y) \in \mathcal{B}_t$ **do**
7:            $A = g(x)$
8:            $A' = \mathrm{TR}_{H,W,K}\big(\mathbb{1}\{|\boldsymbol{\alpha}^c| > Q_\mu\} \otimes \mathbf{J}_{H,W}\big) \odot A$      ▷ Channel-wise sparsity
9:            $\mathcal{M}_t \leftarrow \mathcal{M}_t \cup (A', y)$      ▷ Update episodic memory
10:         **end for**
11:      **end for**
12:      $\boldsymbol{\omega} \leftarrow \mathrm{OPT}_\omega(\omega, A_t)$ as Eq. 7      ▷ Associative memory *write*
13:      **if** $t\ \%\ R == 0$ **then**
14:         Bayesian Training by $\boldsymbol{\omega} \leftarrow \mathrm{OPT}_\omega(\omega)$      ▷ Associative memory *forgetting*
15:      **end if**
16: **end for**

---

memories has been rekindled, and several works have been proposed to achieve strong memory recall performance. However, we notice that how to leverage associative memory in the continual learning setting is under-explored.

In this work, our goal is to design a neuro-inspired continual learning method to improve memory efficiency and mitigate forgetting. To this end, associative memory becomes a natural fit for us due to its properties such as content-based retrieval, fast and efficient recall, and high noise tolerance. Content-based retrieval and noise tolerance allow us to increase the sparsity of the masked feature maps and achieve maximal memory saving. The fast recall reduces the computational overhead for memory retrieval which is critical for our method to be applied to various replay-based baselines.

Formally, an associative memory $\mathcal{A}(\mathbf{x}, \boldsymbol{\omega})$ can be implemented as a recurrent or feed-forward neural network, where $\mathbf{x}$ and $\boldsymbol{\omega}$ denote the input and model parameters of the associative memory, respectively. Corresponding to the "memorize" and "recall" in the human brain, associative memory has *read* and *write* operations which are implemented based on an *energy function*. For example, the predictive-coding-based energy function Rao & Ballard (1999) is the sum of prediction errors across all network layers, i.e.,

$$E\big(\mathbf{x}_{0:L}, \boldsymbol{\omega}_{0:L}\big) = \|\mathbf{x}_L - \boldsymbol{\omega}_L\|_2^2 + \lambda \sum\nolimits_{\ell=0}^{L-1} \|\mathbf{x}_\ell - \mathcal{A}_\ell(\mathbf{x}_{\ell+1}, \boldsymbol{\omega}_\ell)\|_2^2, \tag{5}$$

where $\ell$ is the layer index and $\lambda$ is a coefficient. During training, we write the ground-truth feature map $A$ into associative memory, by minimizing Eq. 5 w.r.t. parameter $\boldsymbol{\omega}$ while keeping the input $\mathbf{x} = A$. During inference, given the masked feature map $A'$ defined in Eq. 4, we retrieve the ground-truth $A$ for memory replay, by minimizing Eq. 5 w.r.t. input $\mathbf{x}$ initialized as $A'$ while keeping the parameter $\boldsymbol{\omega}$ fixed.

## 4.3 TRAINING PIPELINE

Our proposed method involves training the classifier $f_\theta$ and the associative memory $\mathcal{A}$ while maintaining a small episodic memory $\mathcal{M}$. The overall training procedure is shown in Algorithm 1.

**Training classifier.** In each incremental phase, we update the model parameters of the continual learning classifier $f_\theta$ by using the new coming data and the replay samples. The key here is that we use associative memory to retrieve the "complete" feature map and then feed it to the classifier for memory replay. Formally, the training objective of the classifier can be formulated as follows

$$\theta^* = \mathrm{argmin}_\theta \sum\nolimits_{(\mathbf{x},t,\mathbf{y})} \ell\big(f_{\boldsymbol{\theta}}(g(\mathbf{x}), t), \mathbf{y}\big) + \sum\nolimits_{k<t} \ell\big(f_{\boldsymbol{\theta}}, \tilde{\mathcal{M}}_k\big)$$

$$\text{where}\quad \tilde{\mathcal{M}}_k = \mathrm{argmin}_\mathbf{x} E\big(\mathbf{x}_{0:L}, \boldsymbol{\omega}_{0:L}\big) \text{ with } \mathbf{x}_0 \text{ initialized as } A'_k \in \mathcal{M}_k, \tag{6}$$

where the first row is the continual learning objective as defined in Eq. 2. $\tilde{\mathcal{M}}_k$ denotes the retrieved episodic memory, i.e., the feature maps recalled by associative memory. As mentioned earlier, the retrieval by associative memory corresponds to solving an optimization problem as shown in the second row, where $A'_k$ is the masked (before retrieval) feature map from task $k$.

**Training associative memory.** Given feature maps coming from new tasks in each incremental, we need to write those feature maps into the associative memory such that we can ask it to retrieve the complete feature map given a partial cue at a later timestamp. Formally, given feature maps $A_t$ from new task $t$, writing them into associative memory corresponds to solving the following optimization problem

$$\boldsymbol{\omega} = \text{argmin}_{\boldsymbol{\omega}} \ E\big(\mathbf{x}_{0:L}, \boldsymbol{\omega}_{0:L}\big) \text{ with } \mathbf{x}_0 \text{ fixed as } A_t. \tag{7}$$

We also proposed a memory-forgetting mechanism for associative memory to avoid potential memory overload during continual learning. As more data are observed when new tasks keep coming, it is natural for the classifier and associative memory to be biased more towards new tasks' data rather than old ones. To this end, our forgetting mechanism *erases* old tasks' data more times than new ones thus satisfying the inductive bias we want to introduce.

**Update Episodic Memory.** For simplicity, we assume that the memory is populated with the last $m$ examples from each task, although better memory update strategies could be employed (such as building a coreset per task, reservoir sampling, etc.)

## 5 EXPERIMENT

In this section, we evaluate our proposed method SHARC on Class-IL and Task-IL CL. Both performance tables and learning curves over entire tasks are provided. In addition, we demonstrate sensitivity analyses over the masking threshold and comparison of different associative memories. All experiments are conducted on a 64-bit machine with an NVIDIA T4 Tensor Core GPU which has 320 Turing Tensor cores, 2560 CUDA cores, 16GB memory, and Intel® Xeon® Platinum 8259CL CPU @ 2.50GHz. The anonymous code of our method can be found here.

### 5.1 EXPERIMENT SETTING

**Dataset.** In our research, we conducted experiments on three datasets: Split CIFAR-10, Split CIFAR-100, and Split mini-ImageNet Chaudhry et al. (2019). CIFAR-10 consists of 50,000 RGB training images and 10,000 test images, categorized into 10 object classes. Similarly, CIFAR-100 extends this classification task by including 100 classes, with each class containing 600 images. ImageNet-50 comprises 50 classes with 1300 training images and 50 validation images per class. We divided the Split CIFAR-10 dataset into 5 tasks, each with 2 classes. For Split CIFAR-100 and Split mini-ImageNet, we expanded our investigation to 20 tasks, each with 5 classes.

**Comparison Methods.** We compare our method with several replay-based continual learning methods, including: **ER,** a rehearsal-based method that utilizes the average of parameter update gradients from the current task's samples alongside samples from episodic memory to update the learning agent. **MER,** a rehearsal-based model that harnesses the power of episodic memory Riemer et al. (2018). **GEM,** one ensures that valuable information from prior tasks is retained while accommodating new learning Lopez-Paz & Ranzato (2017). **A-GEM,** takes a step further than GEM by incorporating an adaptive mechanism that updates the model's parameters based on both the current task's gradient and the gradients of previous tasks stored in the episodic memory. **CLS-ER,** an innovative algorithm that utilizes a dual-memory learning mechanism to enhance performance in continual learning tasks Arani et al. (2022). **DER++,** a combination of rehearsal, knowledge distillation, and regularization techniques Buzzega et al. (2020).

**Evaluation Metrics.** We assess the classification performance using the ACC metric, which represents the average test classification accuracy across all tasks. We also measure backward transfer (BWT Lopez-Paz & Ranzato (2017)) to evaluate the impact of new learning on previous knowledge. Negative BWT indicates forgetting, so a higher value is preferable. Detailed experimental settings can be found in the appendix.

**Training Details.** We utilized a frozen pre-trained model ImageNet-1K, retaining only the MLP part for training. We directly employed the feature map as the input and output of the associative memory, storing it in a memory buffer. For further details, please refer to the appendix.

Table 1: **Performance comparison on image classification datasets (Task-IL).** The mean and standard deviation are calculated based on five runs with varying seeds. $^+$ denotes the corresponding method combined with our SHARC framework. In most cases, our proposed SHARC framework significantly improves the method.

| Buffer | Model | S-CIFAR-10 | | S-CIFAR-100 | | S-Mini-ImgNet | |
|---|---|---|---|---|---|---|---|
| | | ACC (↑) | BWT (↑) | ACC (↑) | BWT (↑) | ACC (↑) | BWT (↑) |
| - | JOINT | 93.49 ± 0.61 | 43.14 ± 2.07 | 87.57 ± 0.89 | 67.99 ± 1.53 | 74.95 ± 0.7 | 70.02 ± 0.81 |
| - | SGD | 92.31 ± 0.54 | -0.38 ± 0.82 | 85.83 ± 0.35 | 3.08 ± 2.14 | 76.2 ± 0.41 | 3.98 ± 0.75 |
| 200 | GEM | 88.44 ± 1.11 | -4.6 ± 2.24 | 82.82 ± 0.62 | 0.2 ± 1.69 | 72.23 ± 1.26 | -1 ± 1.82 |
| | **GEM$^+$** | 91.01 ± 1.04 | -1.5 ± 1.31 | 83.88 ± 0.52 | -0.04 ± 1.2 | 76.13 ± 0.98 | 3.49 ± 1.47 |
| | A-GEM | 90.52 ± 3.29 | -1.07 ± 1.57 | 85.33 ± 0.58 | 2 ± 1 | 75.18 ± 1.11 | 2.33 ± 1.5 |
| | **A-GEM$^+$** | 91.72 ± 1.02 | -0.35 ± 2.08 | 85.55 ± 0.88 | 1.25 ± 0.63 | 76.54 ± 0.97 | 4.14 ± 1.71 |
| | ER | 86.36 ± 1.33 | -5.04 ± 1.72 | 82.55 ± 0.47 | -0.71 ± 1.59 | 71.66 ± 1.44 | -1.53 ± 2.04 |
| | **ER$^+$** | 91.48 ± 1.18 | -0.93 ± 1.54 | 84.55 ± 0.62 | 0.71 ± 0.61 | 73.68 ± 0.59 | 0.71 ± 0.97 |
| | MER | 87.32 ± 1.39 | -2.3 ± 3.83 | 82.04 ± 0.63 | -0.83 ± 1.49 | 71.2 ± 1.43 | -1.99 ± 1.53 |
| | **MER$^+$** | 91.14 ± 1.62 | -0.9 ± 2.46 | 84.3 ± 0.92 | 0.41 ± 1.26 | 73.54 ± 0.58 | 0.7 ± 1.06 |
| | DER++ | 84.94 ± 1.95 | -6.45 ± 1.91 | 83.27 ± 0.76 | 0.32 ± 1.47 | 72.92 ± 1.09 | -0.13 ± 1.44 |
| | **DER++$^+$** | 89.89 ± 1.34 | -2.72 ± 2.04 | 84.96 ± 0.97 | 0.62 ± 1.44 | 74.59 ± 0.87 | 2 ± 1.27 |
| | CLS-ER | 80.97 ± 2.11 | -12.6 ± 4.39 | 82.97 ± 0.32 | -1.95 ± 1.5 | 73.67 ± 1.05 | -1.75 ± 0.4 |
| | **CLS-ER$^+$** | 91.39 ± 0.7 | -0.94 ± 1.18 | 85 ± 0.41 | 1.27 ± 0.68 | 77 ± 0.45 | 2.7 ± 0.97 |
| 500 | GEM | 88.02 ± 2.61 | -3.84 ± 1.19 | 82.81 ± 0.66 | 0.06 ± 1.66 | 73.6 ± 1.13 | 0.23 ± 1.27 |
| | **GEM$^+$** | 91.53 ± 1.17 | -0.05 ± 1.58 | 84.37 ± 1.03 | 1.63 ± 0.79 | 75.56 ± 0.93 | 3.2 ± 1.61 |
| | A-GEM | 90.81 ± 2.97 | 0.31 ± 2.81 | 85.44 ± 0.28 | 2.32 ± 0.84 | 75.59 ± 1.15 | 2.78 ± 1.61 |
| | **A-GEM$^+$** | 92.32 ± 0.67 | 0.22 ± 0.7 | 85.89 ± 0.82 | 3.25 ± 1.01 | 75.65 ± 0.86 | 3.21 ± 1.56 |
| | ER | 88.05 ± 1.51 | -1.88 ± 3.62 | 82.7 ± 0.63 | -0.09 ± 0.5 | 71.83 ± 1.17 | -1.36 ± 1.5 |
| | **ER$^+$** | 91.64 ± 0.66 | -0.82 ± 0.67 | 84.77 ± 1.57 | 1.85 ± 1.72 | 72.94 ± 0.63 | 0.41 ± 1.08 |
| | MER | 88.33 ± 1.87 | -3.35 ± 1.6 | 82.11 ± 0.5 | -0.36 ± 0.91 | 70.69 ± 1.07 | -2.28 ± 1.41 |
| | **MER$^+$** | 91.69 ± 0.73 | -0.39 ± 1.56 | 84.06 ± 1.33 | 1.43 ± 1.46 | 72.63 ± 0.37 | -0.28 ± 1.08 |
| | DER++ | 86.73 ± 2.77 | -4.79 ± 1.26 | 83.04 ± 0.58 | 0.76 ± 1.64 | 72.05 ± 0.87 | -0.95 ± 1.52 |
| | **DER++$^+$** | 90.46 ± 1.3 | -2.71 ± 1.01 | 85.13 ± 1.57 | 1.81 ± 1.01 | 73.85 ± 0.79 | 1.5 ± 1.5 |
| | CLS-ER | 82.54 ± 3.06 | -8.64 ± 5.52 | 81.34 ± 0.9 | -2.27 ± 1.8 | 72.11 ± 0.38 | -3.41 ± 0.88 |
| | **CLS-ER$^+$** | 90.94 ± 1.49 | -2.22 ± 1.51 | 85.36 ± 0.83 | 1.82 ± 1.66 | 76.27 ± 0.52 | 2.05 ± 0.82 |

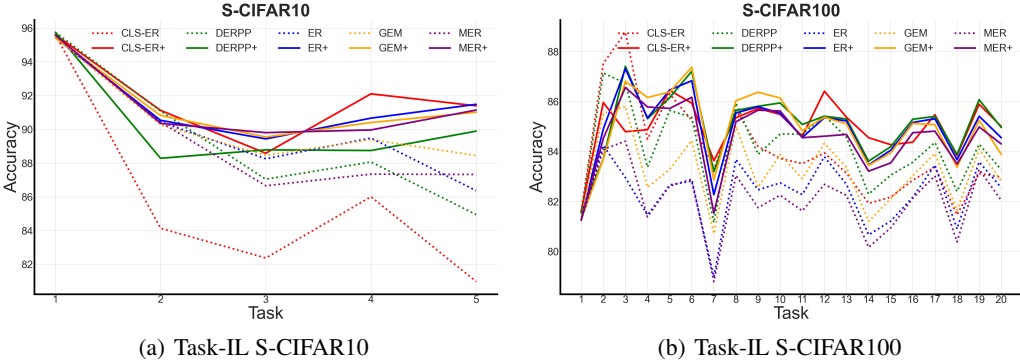

(a) Task-IL S-CIFAR10    (b) Task-IL S-CIFAR100

Figure 3: Learning curves of multiple models with/without SHARC on S-CIFAR10 and S-CIFAR100 in Task-IL scenario. Models with/without SHARC are shown in solid/dotted lines. The buffer size for all models is 200. Methods combined with our proposed framework SHARC significantly prevails.

## 5.2 PERFORMANCE COMPARISON

We demonstrate the impact of our proposed SHARC framework on several state-of-the-art replay-based approaches. Naive baselines such as SGD refer to standard training, while JOINT refers to joint training on all tasks, which provides an upper bound. The experimental results are shown in Table 1 and Table 2, which contain the results in the Task-IL scenario and Class-IL scenario, respectively. Buffer size controls the budget of episodic memory and is distributed evenly to all tasks. Each slot in the buffer contains feature maps of a sample, instead of an image.

Table 1 compares six replay-based methods before and after combining them with SHARC in the Task-IL scenario. Overall, in most cases, the methods used in conjunction with SHARC offer significant improvements. Such contrast exists in all settings (different datasets, models, and buffer sizes). In particular, CLS-ER equipped with SHARC achieves a 12.9% improvement in ACC on

Table 2: **Performance comparison on image classification datasets (Class-IL).** The mean and standard deviation are calculated based on five runs with varying seeds. $^+$ denotes the corresponding method combined with our SHARC framework. In most cases, our proposed SHARC framework significantly improves the method.

| Buffer | Model | S-CIFAR-10 | | S-CIFAR-100 | | S-Mini-ImgNet | |
|---|---|---|---|---|---|---|---|
| | | ACC (↑) | BWT (↑) | ACC (↑) | BWT (↑) | ACC (↑) | BWT (↑) |
| - | JOINT | 72.85 ± 2.18 | 61.26 ± 8.55 | 45.87 ± 1.22 | 45.55 ± 1.45 | 47.08 ± 0.77 | 46.25 ± 0.98 |
| - | SGD | 20.47 ± 0.78 | -90.16 ± 0.92 | 8.55 ± 1.39 | -78.24 ± 0.93 | 12.21 ± 0.75 | -67.11 ± 0.77 |
| 200 | GEM | 27.01 ± 6.16 | -76.88 ± 6.93 | 16.38 ± 3.06 | -67.72 ± 3.84 | 23.76 ± 2.65 | -54.22 ± 3.47 |
| | **GEM$^+$** | 36.39 ± 4.04 | -63.03 ± 7.86 | 20.77 ± 3.22 | -63.08 ± 3.33 | 24.76 ± 1.86 | -52.31 ± 2.62 |
| | A-GEM | 24.12 ± 6.92 | -83.48 ± 3.5 | 12.75 ± 4.2 | -73.96 ± 4.68 | 16.77 ± 2.05 | -62.51 ± 2.71 |
| | **A-GEM$^+$** | 29.19 ± 2.31 | -77.82 ± 3.52 | 17.55 ± 1.91 | -69.61 ± 1.6 | 18.96 ± 1.53 | -60.02 ± 2.11 |
| | ER | 29.35 ± 7.79 | -65.76 ± 10.54 | 14.81 ± 2.75 | -70.15 ± 3.41 | 21.71 ± 1.61 | -56.46 ± 1.88 |
| | **ER$^+$** | 33.94 ± 6.24 | -62 ± 10.91 | 20.58 ± 1.86 | -63.09 ± 1.69 | 23.36 ± 1.76 | -54.24 ± 2.46 |
| | MER | 30.02 ± 7.64 | -61.33 ± 7.94 | 13.74 ± 3.33 | -70.76 ± 3.48 | 21.46 ± 1.78 | -56.59 ± 1.89 |
| | **MER$^+$** | 34.81 ± 5.44 | -59.86 ± 8.47 | 19.46 ± 1.3 | -64.09 ± 1.96 | 22.83 ± 1.88 | -54.68 ± 2.46 |
| | DER++ | 31.55 ± 4.61 | -55.68 ± 10.28 | 14.44 ± 6.09 | -69.28 ± 6.37 | 23.03 ± 1.64 | -54.54 ± 1.74 |
| | **DER++$^+$** | 38.53 ± 4.84 | -39.15 ± 7.8 | 21.01 ± 1.8 | -62.08 ± 2.37 | 24.76 ± 1.3 | -52.16 ± 1.58 |
| | CLS-ER | 27.3 ± 3.13 | -56.75 ± 14.14 | 15.83 ± 1.84 | -70.13 ± 2.49 | 21.77 ± 1.43 | -58.55 ± 2.09 |
| | **CLS-ER$^+$** | 28.63 ± 5.43 | -51.37 ± 8.46 | 18.77 ± 1.93 | -64.2 ± 1.36 | 22.86 ± 1.81 | -57.38 ± 2.31 |
| 500 | GEM | 30.12 ± 10.19 | -63.83 ± 14.88 | 20.81 ± 5.66 | -59.13 ± 6.09 | 30.88 ± 2.39 | -45.43 ± 3.04 |
| | **GEM$^+$** | 33.52 ± 5.29 | -51.17 ± 14.8 | 24.89 ± 1.19 | -54.14 ± 3.14 | 29.8 ± 1.76 | -46.46 ± 2.48 |
| | A-GEM | 25 ± 7.05 | -80.94 ± 2.84 | 12.25 ± 3.87 | -74.35 ± 4.45 | 17.04 ± 0.91 | -62.21 ± 1.11 |
| | **A-GEM$^+$** | 29.21 ± 4.61 | -77.99 ± 5.2 | 15.02 ± 2.02 | -71.12 ± 2.31 | 17.33 ± 1.48 | -61.45 ± 2.16 |
| | ER | 33.16 ± 8.18 | -55.8 ± 12.65 | 19.37 ± 3.9 | -61.39 ± 3.89 | 27.39 ± 1.52 | -49.06 ± 2.18 |
| | **ER$^+$** | 36.18 ± 2.22 | -53.67 ± 6.82 | 24.85 ± 1.41 | -53.85 ± 1.38 | 26.69 ± 0.95 | -49.48 ± 1.61 |
| | MER | 32.96 ± 8.35 | -53.78 ± 14.59 | 20.51 ± 3.97 | -58.51 ± 3.99 | 26.46 ± 1.83 | -49.5 ± 2.45 |
| | **MER$^+$** | 36.04 ± 2.98 | -51.5 ± 2.79 | 23.4 ± 1.52 | -53.89 ± 1.8 | 26.1 ± 1.22 | -50.08 ± 1.35 |
| | DER++ | 34.21 ± 8.62 | -43.95 ± 16.78 | 16.95 ± 5.86 | -63.53 ± 5.67 | 26.48 ± 1.67 | -49.82 ± 2.39 |
| | **DER++$^+$** | 30.57 ± 8.35 | -45.5 ± 8 | 24.66 ± 1.83 | -54.62 ± 2.09 | 25.54 ± 2.78 | -50.72 ± 3.45 |
| | CLS-ER | 26.97 ± 3.32 | -41.42 ± 5.8 | 22.1 ± 2.04 | -58.79 ± 3.31 | 26.74 ± 1.49 | -52.2 ± 1.73 |
| | **CLS-ER$^+$** | 29.64 ± 5.37 | -77.81 ± 5.74 | 22.5 ± 2.52 | -57.04 ± 3.65 | 26.31 ± 1.55 | -52.68 ± 0.55 |

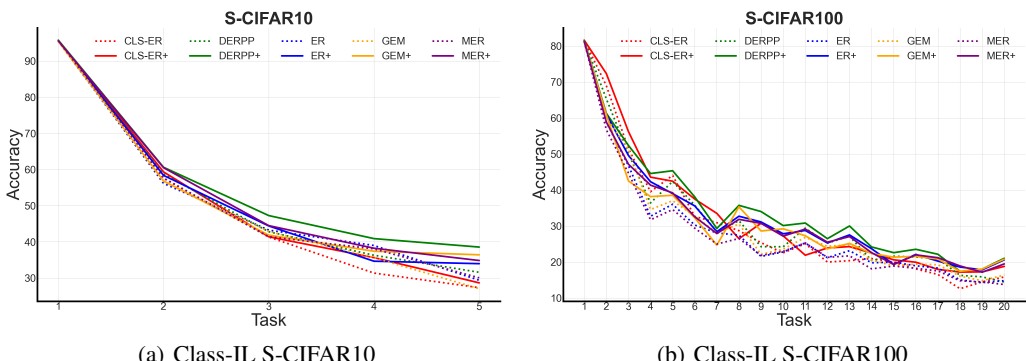

(a) Class-IL S-CIFAR10            (b) Class-IL S-CIFAR100

Figure 4: Learning curves of multiple models with/without SHARC on S-CIFAR10 and S-CIFAR100 in Class-IL scenario. Models with/without SHARC are shown in solid/dotted lines. The buffer size for all models is 200.

S-CIFAR10 with buffer size 200. From a methodological perspective, rehearsal-based methods (e.g., ER) offer greater improvements than constraint-based methods (e.g., GEM). As a typical example, the performance of A-GEM improves only slightly when used with SHARC on S-CIFAR100, which is reasonable since we keep the batch size of the retrieval process constant. Rehearsal-based methods can benefit more from masking because masking reduces the memory space for samples, allowing more previous samples to be reviewed. Furthermore, in most cases on S-CIFAR100 and S-MiniImgNet, the BWT increases or even becomes positive when using SHARC, indicating that SHARC is highly resistant to forgetting. As the buffer size decreases, the complexity of the task increases. Achieving good performance with smaller buffer sizes is the spirit of continual learning. Based on this consideration, we further investigate the learning curve for a minimum buffer size of 200. As shown in Figure 4, methods equipped with SHARC clearly prevail in the figure, indicating that they have been steadily improved during the learning process.

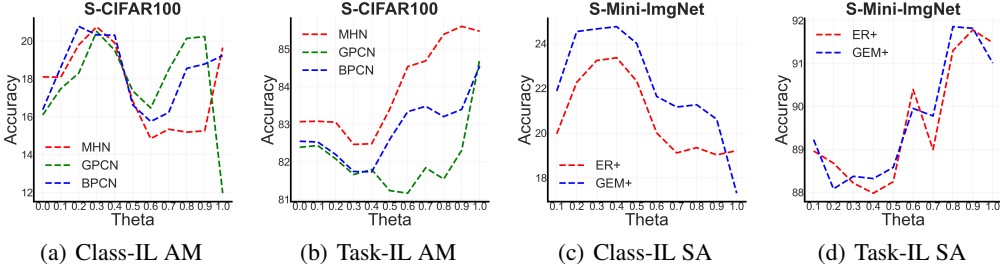

Figure 5: Average accuracy of GEM and ER using different associative memories for inpainting on S-CIFAR100. Figure (a) shows ACC of GEM using different associative memories in Class-IL scenario. Figure (b) shows ACC of ER using different associative memories in Task-IL scenario. The buffer size for all models is 200.

Since the corrupted feature maps after masking cannot be used for backpropagation, in order to keep the backbone network the same as the rest of the work, we froze the convolutional basis of the pre-trained ResNet18, leaving only the parameters of the fully connected layer available for training. A single-layer classifier may not be sufficient in Class-IL, causing all models to perform poorly. Still, this is enough to illustrate the effectiveness of our proposed SHARC framework. Table 2 compares six replay-based methods before and after combining them with SHARC in a Class-IL scenario. Overall, in most cases, the methods used in conjunction with SHARC offer significant improvements. In particular, DER++ equipped with SHARC achieves a 45.5% improvement in ACC on S-CIFAR100. The smaller the buffer, the more pronounced this contrast becomes. In particular, with a buffer size of 200, SHARC improves CLS-ER in ACC much greater compared to the buffer size of 500 on S-CIFAR100.

### 5.3 ASSOCIATIVE MEMORY COMPARISON

As shown in Figure 5(a) and Figure 5(b), we compare different configurations of the associative memory. We in general follow Yoo & Wood (2022) for the implementation. We found that Modern Hopfield Network Ramsauer et al. (2020) favors more towards the task-incremental setting while BayesPCN Yoo & Wood (2022) favors more towards class-incremental setting. This is potentially due to that, in BayesPCN a forgetting mechanism is implemented, which can help mitigate the memory overload of the associative memory when the samples to memorize are too many. Such design seems to play a more important role in the more challenging class-incremental setting.

### 5.4 MASKING THRESHOLD SENSITIVITY ANALYSIS

We conduct sensitivity analysis on the threshold $Q_\mu$ in Eq 4. Masking threshold $Q_\mu$ is defined as certain percentile value for 512 feature importance. Features with importance below the threshold will be masked. $Q_\mu$ is controlled by a hyper-parameter Theta. Theta ranges between (0, 1], where 1 means that only the features with the highest importance are retained. While different methods may favor different optimal thresholds, a general sensitivity analysis is still helpful in determining optimal threshold settings. As shown in Figure 5(d), the optimal Theta in Task-IL is between 0.8 to 0.9. As shown in Figure 5(c), the optimal Theta in Class-IL is between 0.3 to 0.5. This indicates that the optimal thresholds show different trends in Task-IL and Class-IL. In particular, Task-IL requires a higher threshold to drop most features, while the opposite is true for Class-IL. This is reasonable because in Task-IL, the task ID is given as additional information. Whereas in Class-IL, the model can only get additional information from more features.

## 6 CONCLUSION

We propose SHARC, a novel framework that bridges the gap between current AI models and humans in continual learning. Combining associative memory and interpretive techniques, SHARC enables efficient, near-perfect recall of seen samples in a human-like manner. As a generic framework, SHARC can be seamlessly adapted to any replay-based approach, thus improving their performance in different continual learning scenarios. We demonstrate the effectiveness of our framework with abundant experimental results. Our proposed SHARC framework consistently improves several SOTA replay-based methods on multiple benchmark datasets.

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

# A APPENDIX

## A.1 EXPERIMENTAL DETAILS

### A.1.1 DATASET DETAILS

We expand upon the datasets used for our experiments in this section. We highlighted the sentence that describes the domain drift within each dataset.

- **CIFAR-10:** The CIFAR-10 dataset is a comprehensive collection of $60,000$ $32 \times 32$ color images divided into 10 distinct classes, with $6,000$ images per class. This dataset is further split into $50,000$ training images and $10,000$ test images, allowing for effective model evaluation.

- **CIFAR-100:** The dataset is similar to CIFAR-10 and is composed of 100 classes, each containing 600 images. Specifically, within the 600 images, there are 500 images used for training and 100 images designated for testing purposes. It is important to note that the 100 classes are actually comprised of 20 classes, where each class is further divided into 5 sub-classes. Therefore, the total count of 100 classes is obtained by multiplying 5 and 20 ($5 \times 20 = 100$).

- **Mini-ImageNet:** The Mini-ImageNet dataset contains 100 classes with a total of $60,000$ color images. Each class has 600 samples, and the size of each image is $84 \times 84$ pixels. Typically, the class distribution between the training and testing sets of this dataset is $80 : 20$. Compared to the CIFAR-10 dataset, the Mini-ImageNet dataset is more complex but is better suited for prototype design and experimental research. Chaudhry et al. (2019)

### A.1.2 DETAILS OF COMPARISON METHODS

In this paper, we compare our proposed SHARC with several SOTA replay-based methods as well as regularization-based methods. Specifically,

- **ER,** a rehearsal-based method that utilizes the average of parameter update gradients from the current task's samples alongside samples from episodic memory to update the learning agent. This method, known as ER (Episodic Regularization), offers a computationally efficient alternative to GEM (Gradient Episodic Memory) and has demonstrated successful performance when dealing with a limited memory buffer.

- **MER,** a rehearsal-based model that harnesses the power of an episodic memory. MER employs a unique loss function that approximates the dot products of the gradients of current and previous tasks, thereby mitigating the issue of forgetting. To ensure a fair and comprehensive comparison with other methods, we adjust the experimental setting by setting the number of inner gradient steps to 1 for each outer meta-update, while maintaining a mini-batch size of 10. This adjustment allows us to establish a more consistent framework for evaluating the performance of MER alongside other approaches, specifically in terms of the number of stochastic gradient descent (SGD) updates. By presenting these findings, we aim to shed light on the effectiveness and practicality of MER as a rehearsal-based model in the context of meta-learning Riemer et al. (2018).

- **GEM,** who utilizes an episodic memory buffer to store past experiences and gradients. By incorporating both the current task's gradient and the gradients of previous tasks from the episodic memory, GEM ensures that valuable information from prior tasks is retained while accommodating new learning. To prevent catastrophic forgetting, the algorithm employs a constrained optimization approach, projecting the current gradient onto a subspace that preserves knowledge from previous tasks Lopez-Paz & Ranzato (2017).

- **A-GEM,** takes a step further than GEM by incorporating an adaptive mechanism that updates the model's parameters based on both the current task's gradient and the gradients of previous tasks stored in the episodic memory. This allows AGEM to effectively preserve knowledge from prior tasks while adapting to new tasks.

- **CLS-ER,** an innovative algorithm that utilizes a dual-memory learning mechanism to enhance performance in continual learning tasks. In this approach, the episodic memory serves as a repository for storing samples encountered during the learning process. On the other hand, semantic memories play a crucial role in constructing short-term and long-term memories of the learned representations from the working model Arani et al. (2022).

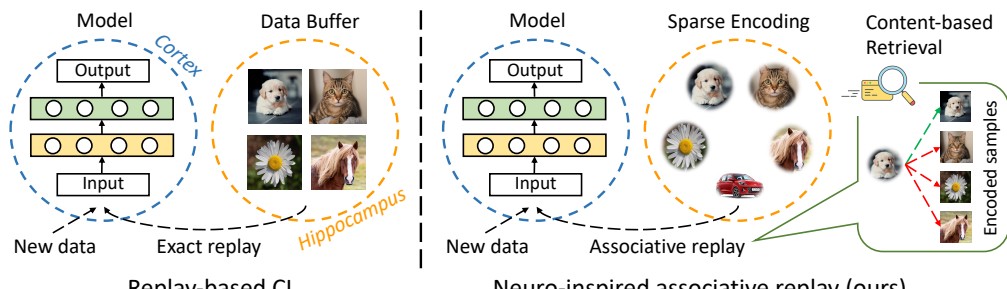

Figure 6: Motivation and overview of our proposed framework. **Left:** Typical replay-based approaches maintain and replay a small episodic memory of previous samples, which is inspired by the cortex and hippocampus in the human brain. **Right:** In FaR, our memory buffer is equipped with a forgetting mechanism to drop uninformative episodes, and a content-addressable associative memory is used to achieve fast and high-accuracy data retrieval.

- **DER++,** a combination of rehearsal, knowledge distillation, and regularization techniques. This approach leverages the network's logits sampled at different stages of the optimization trajectory. This approach promotes consistency with the network's past experiences Buzzega et al. (2020).

### A.1.3 HYPER-PARAMETER SETTING

All experiments are conducted on a 64-bit machine with an NVIDIA T4 Tensor Core GPU which has 320 Turing Tensor cores, 2560 CUDA cores, 16GB memory, and Intel® Xeon® Platinum 8259CL CPU @ 2.50GHz. The learning rate for all datasets is uniformly set to be 0.1. Down below we report the hyper-parameter unique to some models applied in our experiment.

- **ER:**
  'lr': 0.1

- **MER:**
  'lr': 0.1, 'gamma': 0.5, 'batch num': 1

- **GEM:**
  'lr': 0.1, 'gamma': 0.5

- **AGEM:**
  'lr': 0.1

- **DER++:**
  'lr': 0.1, 'alpha': 0.1, 'beta': 0.5

- **CLSER:**
  'lr': 0.1, 'reg weight': 0.15, 'stable model update freq': 0.1, 'stable model alpha': 0.999, 'plastic model update freq': 0.3, 'plastic model alpha': 0.999

### A.2 ADDITIONAL PRELIMINARIES

In this section, we provide more discussion about some preliminaries in this paper.

### A.3 MOTIVATION OF SHARC

#### A.3.1 ASSOCIATIVE MEMORY

When applied to Computer Science problems, associative memories come in two high-level forms: *auto-associative* and *hetero-associative* memories. While both are able to recall patterns given a set of inputs, auto-associative memories are primarily focused on recalling a pattern X when provided a partial or noisy variant of X. By contrast, hetero-associative memories are able to recall not only patterns of different sizes from their inputs but can be leveraged to map concepts between categories (hence "hetero-associative"). One common example from the literature is a hetero-associative memory that might recall the embedded animal concept of "monkey" given the embedded food concept of "banana". Since all forms of AM are focused on the actual content being stored and retrieved, they are also commonly referred to as *content-addressable memories* (CAM) in the literature.

**Classical Hopfield Network.** One of the earliest and probably the most well-known auto-associative memory are Hopfield Networks Hopfield (1982). The original Hopfield Networks are discrete where they operate by storing binary-pattern inputs into the weights of a fully-connected neural network using a local update rule. For an input $\mathbf{x} \in \{-1, 1\}^d$ containing $d$ binary values, a Hopfield Network contains $d^2$ real-valued connections, i.e., $\mathbf{W} \in \mathbb{R}^{d \times d}$. The Hopfield learning algorithm specifies a write over $n$ binary memories $\mathbf{x}_1, \mathbf{x}_2, \cdots, \mathbf{x}_n$, represented as column vectors, by accumulating their outer product

$$\mathbf{W} = \sum_{i=1}^{n} \mathbf{x}_i \mathbf{x}_i^{\mathsf{T}}, \quad \mathbf{W}[p, q] = \sum_{i=1}^{n} \mathbf{x}_i[p] \mathbf{x}_i^{\mathsf{T}}[q]. \tag{8}$$

This specific write update is termed the *Hebbian update rule* as it follows the "fire together, wire together" principle proposed by psychologist Donald Hebb as a model of synaptic learning Hebb (1949). The Hopfield Network *read* operation involves minimizing an *energy function*

$$E(\mathbf{W}, \boldsymbol{\xi}) = -\frac{1}{2} \boldsymbol{\xi}^{\mathsf{T}} \mathbf{W} \boldsymbol{\xi}, \tag{9}$$

where $\boldsymbol{\xi} \in \mathbb{R}^d$ is the *state* of the network, initialized as the initial query, and then optimized to a stable state known as the *attractor*.

**Modern Hopfield Network.** Hopfield Networks serve as an interesting weight-based method of memory storage. However, although they use optimization as a method of memory retrieval, their learning rule is not differentiable due to the use of discrete states. Modern Hopfield Network (MHN Ramsauer et al. (2020)) introduces a new energy function instead of that in Eq. 9. Specifically, MHN generalizes the energy function to continuous-valued patterns and adds a quadratic term, i.e.,

$$E(\mathbf{X}, \boldsymbol{\xi}, \beta) = -\text{LSE}(\beta, \mathbf{X}^{\mathsf{T}} \boldsymbol{\xi}) + \frac{1}{2} \boldsymbol{\xi}^{\mathsf{T}} \boldsymbol{\xi} + \beta^{-1} \log(N) + \frac{1}{2} M^2, \tag{10}$$

where $\mathbf{X}$ is the matrix form of $N$ *continuous* stored patterns $\mathbf{x}_i$, $i = 1, 2, \cdots, N$, $M$ is the largest norm of all stored patterns, LSE stands for the LogSumExp function with coefficient $\beta$.

**Predictive Coding Network.** The Predictive Coding Network is a computational model that aims to explain how the brain processes sensory information and makes predictions about future sensory inputs. It is based on the concept of predictive coding, which suggests that the brain constantly generates predictions about upcoming sensory inputs and updates these predictions based on the actual sensory feedback it receives. In machine learning, the predictive coding network is implemented as an energy-based associative memory model that has set the state-of-the-art on a number of image associative recall tasks.

## A.4 Additional Experimental Results

Table 3: **Performance comparison on image classification datasets (Task-IL).** The mean and standard deviation are calculated based on five runs with varying seeds. $^+$ denotes the corresponding method combined with our SHARC framework.

| Buffer | Model | S-CIFAR-10 | | S-CIFAR-100 | | S-Mini-ImgNet | |
|---|---|---|---|---|---|---|---|
| | | ACC (↑) | BWT (↑) | ACC (↑) | BWT (↑) | ACC (↑) | BWT (↑) |
| - | JOINT | 72.85 ± 2.18 | 61.26 ± 8.55 | 45.87 ± 1.22 | 45.55 ± 1.45 | 47.08 ± 0.77 | 46.25 ± 0.98 |
| - | SGD | 20.47 ± 0.78 | -90.16 ± 0.92 | 8.55 ± 1.39 | -78.24 ± 0.93 | 12.21 ± 0.75 | -67.11 ±0.77 |
| 1000 | GEM | 89.07 ± 1.09 | -3.28 ± 1.61 | 82.38 ± 0.93 | 2.59 ± 1.08 | 73.78 ± 1 | 0.38 ± 1.53 |
| | **GEM**$^+$ | 92.14 ± 0.39 | -0.66 ± 1.93 | 85.56 ± 0.76 | 1.97 ± 1.8 | 75.11 ± 0.59 | 3.69 ± 1.79 |
| | A-GEM | 91.01 ± 2.86 | -0.46 ± 1.09 | 85.7 ± 0.45 | 2.53 ± 1.43 | 75.86 ± 0.92 | 3.02 ± 1.38 |
| | **A-GEM**$^+$ | 92.13 ± 0.49 | -0.44 ± 2.05 | 86.67 ± 0.87 | 3.46 ± 1.45 | 75.42 ± 0.88 | 4.02 ± 1.74 |
| | ER | 89.56 ± 1.09 | -1.33 ± 3.47 | 83.58 ± 1.12 | 1.39 ± 1.17 | 71.45 ± 1.01 | -1.71 ± 1.37 |
| | **ER**$^+$ | 91.81 ± 0.25 | -0.83 ± 1.82 | 85.71 ± 0.63 | 2.34 ± 1.2 | 74.18 ± 0.82 | 2.53 ± 1.54 |
| | MER | 89.05 ± 2.37 | -2.82 ± 3.1 | 82.5 ± 1.05 | 0.1 ± 2.23 | 70.17 ± 1.22 | -3.16 ± 1.66 |
| | **MER**$^+$ | 91.7 ± 0.55 | -0.78 ± 1.01 | 85.08 ± 1.19 | 1.53 ± 1.06 | 72.37 ± 0.87 | 0.29 ± 1.7 |
| | DER++ | 87.56 ± 1.87 | -2.14 ± 4.63 | 83.57 ± 0.64 | 1.62 ± 1.31 | 71.83 ± 1.12 | -0.98 ± 1.46 |
| | **DER++**$^+$ | 90.4 ± 0.63 | -3.1 ± 0.77 | 85.87 ± 0.92 | 2.54 ± 1.22 | 74.09 ± 0.85 | 2.82 ± 2 |
| | CLS-ER | 83.02 ± 2.81 | -8.88 ± 4.33 | 82.67 ± 0.73 | -0.28 ± 1.5 | 71.97 ± 0.43 | -3.69 ± 1.02 |
| | **CLS-ER**$^+$ | 89.81 ± 1.77 | 0.85 ± 3.75 | 85.17 ± 0.61 | 1.24 ± 2.02 | 76.35 ± 0.41 | 1.97 ± 0.52 |

In general, when combined with SHARC, the methods used show notable enhancements in most scenarios. Specifically, in the experiments conducted with a buffer size of 1000 and CIFAR-10, the maximum improvement in accuracy reaches approximately 7%.

Table 4: **Performance comparison on image classification datasets (Class-IL).** The mean and standard deviation are calculated based on five runs with varying seeds. $^+$ denotes the corresponding method combined with our SHARC framework.

| Buffer | Model | S-CIFAR-10 | | S-CIFAR-100 | | S-Mini-ImgNet | |
|---|---|---|---|---|---|---|---|
| | | ACC (↑) | BWT (↑) | ACC (↑) | BWT (↑) | ACC (↑) | BWT (↑) |
| - | JOINT | 72.85 ± 2.18 | 61.26 ± 8.55 | 45.87 ± 1.22 | 45.55 ± 1.45 | 47.08 ± 0.77 | 46.25 ± 0.98 |
| - | SGD | 20.47 ± 0.78 | -90.16 ± 0.92 | 8.55 ± 1.39 | -78.24 ± 0.93 | 12.21 ± 0.75 | -67.11 ±0.77 |
| 1000 | GEM | 32.13 ± 6.79 | -56.21 ± 13.61 | 23.39 ± 5.47 | -48.35 ± 5.51 | 31.92 ± 5.29 | -42.6 ± 6.71 |
| | **GEM**$^+$ | 40.68 ± 3.18 | -49.08 ± 11.47 | 27.49 ± 3.49 | -44.72 ± 3.92 | 33.54 ± 2.24 | -40.21 ± 4.06 |
| | A-GEM | 26.2 ± 8.53 | -80.68 ± 7.31 | 13.38 ± 3.81 | -73.22 ± 4.95 | 16.52 ± 1.54 | -62.86 ± 2.1 |
| | **A-GEM**$^+$ | 28.93 ± 3.67 | -78.13 ± 5.81 | 16.04 ± 3.47 | -70.52 ± 3.49 | 17.58 ± 1.71 | -60.32 ± 1.7 |
| | ER | 33.82 ± 8.42 | -53.69 ± 15.06 | 23.13 ± 5.06 | -53.35 ± 5.2 | 30.77 ± 1.85 | -43.9 ± 2.47 |
| | **ER**$^+$ | 37.74 ± 2.45 | -43.36 ± 14.88 | 26.75 ± 1.1 | -49.72 ± 0.86 | 30.26 ± 1.95 | -43.64 ± 1.07 |
| | MER | 32.87 ± 8.8 | -50.42 ± 18.37 | 23.59 ± 3.26 | -51.08 ± 4.59 | 29.73 ± 2.36 | -44.68 ± 2.92 |
| | **MER**$^+$ | 36.94 ± 5.38 | -44.32 ± 14.72 | 25.22 ± 0.8 | -49.68 ± 1.81 | 29.71 ± 1.37 | -43.58 ± 1.47 |
| | DER++ | 31.37 ± 7.22 | -52.59 ± 16.29 | 19.43 ± 7.88 | -59.2 ± 7.87 | 29.33 ± 1.83 | -45.58 ± 2.29 |
| | **DER++**$^+$ | 35.51 ± 7.44 | -36.06 ± 11.08 | 26.35 ± 1.84 | -52.05 ± 2.85 | 29.42 ± 2.02 | -44.78 ± 2.61 |
| | CLS-ER | 24.13 ± 8.02 | -33.43 ± 23.68 | 25.71 ± 2.91 | -51.68 ± 2.84 | 30.3 ± 1.51 | -47.22 ± 1.7 |
| | **CLS-ER**$^+$ | 26.34 ± 5.38 | -43.81 ± 7.09 | 23.99 ± 4.36 | -53.42 ± 5.22 | 30.21 ± 1.25 | -46.88 ± 1.73 |

Our method demonstrates strong performance on the image classification dataset task, with improvements observed across various metrics compared to the original method. Notably, even with a large buffer, we achieve an average improvement of approximately 3% in accuracy, providing compelling evidence of the effectiveness of our approach.

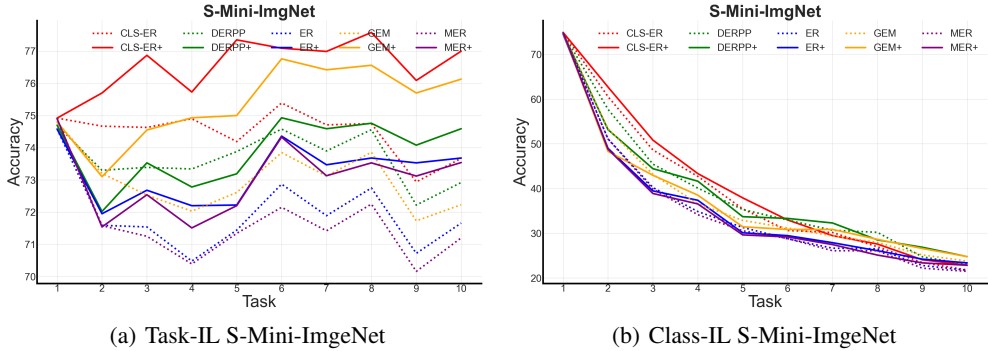

(a) Task-IL S-Mini-ImgeNet

(b) Class-IL S-Mini-ImgeNet

Figure 7: Learning curves of mumltiple models with / without SHARC on S-Mini-ImgeNet. Models with / without SHARC are shown in solid / dotted lines. Buffer size for all models is 200.

It is evident that our approach significantly enhances the model across multiple dimensions. Notably, in terms of task accuracy, all algorithms show an improvement of approximately 2%, with this value consistently increasing as the number of tasks grows. Particularly noteworthy is the observation that while the accuracy of the original algorithms tends to decrease with larger tasks, our method continues to increase in accuracy, demonstrating its effectiveness in handling a large number of multi-tasks. These findings strongly indicate the superior performance of our method in scenarios involving numerous tasks.

