# OpenReview forum: "Saliency-Guided Hidden Associative Replay for Continual Learning"
_ICLR.cc/2024/Conference — Submitted to ICLR 2024_

### Official Review · Reviewer_8orT · 2023-10-24

**Soundness:** 2 fair
**Presentation:** 2 fair
**Contribution:** 2 fair
**Rating:** 3
**Confidence:** 4

**Summary:**

This paper introduces Saliency-Guided Hidden Associative Replay for Continual Learning (SHARC), a method that attempts to tackle the catastrophic forgetting problem in continual learning by incorporating associative memory into replay-based strategies. The memory selectively stores essential feature map channels determined using saliency methods like Grad-CAM. Experiments on several benchmarks show improvements over some existing replay-based CL methods under task-incremental and class-incremental learning settings.

**Strengths:**

- The paper's endeavor to draw inspiration from insights into human brain function for addressing CL is commendable.

- The proposed method improves several existing replay-based CL methods.

**Weaknesses:**

1. The working mechanism of associative memory is confusing.
- It is unclear how the queries are obtained during both training and inference.
- The concept of queries, which appears to be image crops in Fig. 2, contradicts the notion presented in the paper, where specific feature map channels are retained.
- The practicality of recalling precisely the same image when given a query is questionable, as queries pertain to new classes, while the memory contents relate to old classes. This discrepancy, unless old queries are also stored, fundamentally undermines the approach. Storing old queries, however, introduces extra complexities that appear inconsistent with the intended advantages of using associative memory.


2. The reliance on gradient-based saliency methods to evaluate the importance of feature map channels may not align with the underlying feature selection mechanisms in the human brain.


3. The related work is noticeably lacking in depth.
- The paper lacks a comprehensive overview of both regularization-based and dynamic architecture-based continual learning methods. Notably, there is a complete absence of references to the latter category.
- The review neglects to consider replay-based continual learning methods that store lightweight features or generate pseudo features for old classes [1-3], which should be addressed, discussed, and compared.
- The absence of a thorough examination of other continual learning methods employing brain-inspired memory systems, such as [4-5], leaves a critical gap that requires addressing through comprehensive review, discussion, and comparison.


4. In terms of comparisons:
- It is unclear whether methods without SHARC in Tables I and II are also pre-trained on ImageNet.
- The authors are encouraged to extend the comparison to include more recent methods introduced in 2022 or 2023.


5. The claim that "existing work Sun et al. (2015) has proved that hidden representation learned by convolutional neural networks is highly sparse in the hidden space" is inadequately supported, as the referenced work by Sun et al. (2015) predominantly focuses on face representations rather than general representations acquired through CNNs, rendering the argument questionable.


Reference

[1] Generative feature replay for class-incremental learning. CVPRW 2020.

[2] Self-sustaining representation expansion for non-exemplar class-incremental learning. CVPR 2022.

[3] Fetril: Feature translation for exemplar-free class-incremental learning. WACV 2023.

[4] Learning Fast, Learning Slow: A General Continual Learning Method based on Complementary Learning System. ICLR 2022

[5] Sparse Coding in a Dual Memory System for Lifelong Learning. AAAI 2023

**Questions:**

- What is the value of K? Is it the same as the number of classes in order to perform the Grad-CAM? Will K change under the class-incremental setting?

- What are the two multiplication operations in Eq. (4), respectively?

- Is A’ a feature map after discarding the non-salient channels?

- Can you provide more explanations for the sentence “We only need to keep track of the channel index which is only a 1d vector and cheap to store”?

- What is the “partial cue” that is used to retrieve feature maps in the associative memory?

---

> ### Author Response · Authors · 2023-11-23
> **Related Work and Comparison(Part1)**
>
> Thank you for your time and dedication to this review process. We are grateful for the opportunity to benefit from your expertise, and we believe your contributions will enhance the overall quality of the SHARC.
>
> >*"The paper lacks a comprehensive overview of both regularization-based and dynamic architecture-based continual learning methods. Notably, there is a complete absence of references to the latter category."*
>
> **A1.**
> Architect-based:
>
> * DER: Dynamically Expandable Representation for Class Incremental Learning CVPR 2021 \
> * Progressive Neural Networks 2016 \
> * Lifelong Learning with Dynamically Expandable Networks 2017 \
> * Progress & Compress: A scalable framework for continual learning PMLR 2018 \
> * DyTox: Transformers for Continual Learning With DYnamic TOken eXpansion CVPR 2022
>
> Regularization-based:
>
> * Uncertainty-based Continual Learning with Adaptive Regularization NerulPS 2019 \
> * CPR: Classifier-Projection Regularization for Continual Learning ICLR 2020 \
> * Adversarial Targeted Forgetting in Regularization and Generative-Based Continual Learning Models IJCNN 2021 \
> * Continually Learning Self-Supervised Representations with Projected Functional Regularization CVPRW 2022
>
> >*"The absence of a thorough examination of other continual learning methods employing brain-inspired memory systems, such as [4-5], leaves a critical gap that requires addressing through comprehensive review, discussion, and comparison."*
>
> **A2.**
> [4] CLSER \
> The method described in quote [4], called CLSER, has already been extensively discussed in this paper. We applied our own method, known as SHARC, to the CLSER framework and conducted evaluations on three datasets: CIFAR-10, CIFAR-100, and TinyImageNet. The results clearly demonstrate a significant improvement in performance, particularly when the buffer size is relatively smaller.
>
> [5] SCoMMER \
> When comparing our method to CLSER, we observe that our method achieves similar results to SCoMMER on the CIFAR-100 dataset. However, our method surpasses SCoMMER by one percentage point in accuracy when the buffer size is set to 200. Conversely, SCoMMER underperforms CLSER on the CIFAR-10 dataset. In contrast, our SHARC method achieves an even higher accuracy, approximately three percentage points higher than CLSER, particularly when the buffer size is 500. This stark contrast fully illustrates the robustness of our method across different datasets and underscores its superiority.
>
> >*"It is unclear whether methods without SHARC in Tables I and II are also pre-trained on ImageNet."*
>
> **A3.**
> In our approach, SHARC, we adopt a consistent starting point by replacing the original network structure, such as ER, MER or GEM, with a trained and partially parameterized ResNet. This ResNet serves as the backbone for processing all image data in our framework.
>
> One notable advantage of SHARC is its flexibility in accommodating different methods. Regardless of the specific method employed in combination with SHARC, the main differentiating factors lie in the fixed structure of the original methods and their respective optimal training parameters. By adhering to this consistent framework, we ensure that the primary distinctions among the methods stem from their unique configurations and parameter choices.
>
> This approach allows for a fair and meaningful comparison of different methods within the SHARC framework, as the focus shifts toward analyzing the impact of their specific configurations and parameter settings on overall performance. By isolating these factors, we can better understand the strengths and weaknesses of each method and make informed decisions when applying them to different datasets or problem domains.

---

### Official Review · Reviewer_vmux · 2023-10-27

**Soundness:** 3 good
**Presentation:** 4 excellent
**Contribution:** 2 fair
**Rating:** 3
**Confidence:** 3

**Summary:**

Paper proposes a novel Continual Learning (CL) method, SHARC, inspired by biological memory (selective memory retention of salient experience). It has 2 novel contributions to the CL problem: saliency selection of feature channels and Associative Memory (AM) to improve memory efficiency and mitigate forgetting. The empirical experiments for the Task-IL and the Class-IL settings show significant improvement over the SOTA replay CL methods like GEM, A-GEM, ER, etc.

**Strengths:**

Paper's position is that the biological inspired architecture for CL is more memory efficient and can outperform the current replay methods which either stores the entire representation or approximately generate the training data for older classes. This motivation is supported by the strong experimental results.

**Weaknesses:**

1. The novelty of the paper is limited as it combines existing methods such as "saliency" and "associative memory" for the CL problem. In image retrieval literature, there are several prior works which combine both mechanisms. See references below. As such, the real novelty lies only in the application of these 2 mechanism to the CL problem.

2. (Minor) Experiments only include replay-based CL methods. As mentioned in the paper's related works, replay methods are among the strongest in CL. However, this design leaves a big unknown about how well the proposed method compare against other approaches, like regularization-based and dynamic architecture-based. Only DER++ which has regularization and rehearsal is included.

3. (minor) The implementation details about how SHARC combine with the other baseline methods are not given. However, this is somewhat mitigated by the inclusion of anonymous codes. I did not inspect the codes, however.

Ge, S. S., Li, M., & Lee, T. H. (2016). Dynamic saliency-driven associative memories based on network potential field. Pattern Recognition, 60, 669-680.
Kuo, D. W., Cheng, G. Y., Cheng, S. C., & Lee, S. L. (2012, October). Detecting salient fragments for video human action detection and recognition using an associative memory. In 2012 International Symposium on Communications and Information Technologies (ISCIT) (pp. 1039-1044). IEEE.

REVISED: See major weakness from Reviewer WbsA.

**Questions:**

1. (Section 5.2) How does the proposed method combine with the 6 replay-based methods? Does the SHARC Memory Replay module directly replace the respective methods' original replay mechanism? What about the SHARC prediction head? How does it combine with the other methods?

2. By "combining" SHARC with other methods, what's the network size increased? In general, we can expect improved performance for many tasks simply by increasing the network size. So it's important to know the increase of network size, after combining with SHARC.

**Details Of Ethics Concerns:**

No concerns

---

> ### Author Response · Authors · 2023-11-23
> **Implementation Details(Part1)**
>
> Thank you for your time and dedication to this review process. We are grateful for the opportunity to benefit from your expertise, and we believe your contributions will enhance the overall quality of the SHARC.
>
> >*"The implementation details about how SHARC combines with the other baseline methods are not given. However, this is somewhat mitigated by the inclusion of anonymous codes. I did not inspect the codes, however."*
>
> **A1.**
> First, we start by replacing the backbone in the original method with a pre-trained and parameter-frozen ResNet. This allows us to leverage the pre-trained weights and architecture of ResNet, which is a popular convolutional neural network. By doing this, we obtain the feature map from the modified ResNet.
>
> Next, we incorporate a simple MLP as the prediction head. The MLP takes the feature map as input and performs the task of making predictions based on the extracted features.
>
> When a batch of data is entered into the system, the first step is to obtain the feature importance using our modified Grad-CAM which is a technique that highlights the important regions in an image that contribute to the predictions made by the model. By applying Grad-CAM on the modified ResNet, we obtain a feature map based on the input data.
>
> After obtaining the feature map, we update the gradients on the current minibatch using this modified Grad-CAM. This step helps us to fine-tune the model based on the specific data in the current batch.
>
> Next, we utilize Grad-CAM masks to mask the feature maps of the images within the batch. Grad-CAM masks allow us to focus on the most relevant parts of the feature maps, enabling us to extract more meaningful information.
>
> Once the feature maps are masked, we process and recover them using BayesPCN. Finally, the recovered feature maps are stored in the episodic memory, which serves as a repository for important information that can be later accessed and utilized for various purposes.
>
> >*"How does the proposed method combine with the 6 replay-based methods? Does the SHARC Memory Replay module directly replace the respective methods' original replay mechanism? What about the SHARC prediction head? How does it combine with the other methods?"*
>
> **A2.**
> First, we start by replacing the backbone in the original method with a pretrained and parameter-frozen ResNet. This allows us to leverage the pre-trained weights and architecture of ResNet, which is a popular convolutional neural network. By doing this, we obtain the feature map from the modified ResNet.
>
> Next, we incorporate a simple MLP as the prediction head. The MLP takes the feature map as input and performs the task of making predictions based on the extracted features.
>
> When a batch of data is entered into the system, the first step is to obtain the feature importance using our modified Grad-CAM which is a technique that highlights the important regions in an image that contribute to the predictions made by the model. By applying Grad-CAM on the modified ResNet, we obtain a feature map based on the input data.
>
> After obtaining the feature map, we update the gradients on the current minibatch using this modified Grad-CAM. This step helps us to fine-tune the model based on the specific data in the current batch.
>
> Next, we utilize Grad-CAM masks to mask the feature maps of the images within the batch. Grad-CAM masks allow us to focus on the most relevant parts of the feature maps, enabling us to extract more meaningful information.
>
> Once the feature maps are masked, we process and recover them using BayesPCN. Finally, the recovered feature maps are stored in the episodic memory, which serves as a repository for important information that can be later accessed and utilized for various purposes.
>
> In summary, we made several modifications to the original methods. Firstly, we replaced the backbone of the network with a different architecture. This change allows us to leverage the strengths of the new backbone and potentially improve the performance of the network. Additionally, we modified the form of the data in the episodic memory. This alteration enables us to store and retrieve information in a more efficient and effective manner. By adapting the data structure to better suit our needs, we can enhance the overall performance of the network. Furthermore, we introduced BayesPCN as the associative memory network to achieve strong memory recall performance.
>
> Overall, these changes contribute to improving the performance and memory capabilities of the network, allowing it to handle complex tasks better and make more accurate predictions.

---

> ### Author Response · Authors · 2023-11-23
> **Implementation Details(Part2)**
>
> >*"By "combining" SHARC with other methods, what's the network size increased? In general, we can expect improved performance for many tasks simply by increasing the network size. So it's important to know the increase of network size, after combining with SHARC."*
>
> **A3.**
> Take the method "erbpcn" as an example. In the specific case of theta=0.2, where only 20% of the original image is retained, if we were to deposit all the complete images inside the episodic memory, it would require approximately 7.5 million parameters. However, with our SHARK method, the episodic memory is represented by a single feature map, resulting in a reduction to only about 1.3 million parameters. This optimization effectively drops about 6 million parameters, and the computation involved is only 17% of the original.
>
> Comparatively, the ResNet18 architecture, which is commonly used as a backbone in deep learning models, consists of around 11 million parameters. In contrast, the associative memory network that we introduced contains only about 4 million parameters. By adopting our approach, we achieve a memory consumption reduction of approximately 11% compared to the original tasker, while also obtaining higher accuracy.
>
> These results clearly demonstrate the efficiency and accuracy of our method. By optimizing the episodic memory representation and introducing the associative memory network, we significantly reduce the parameter count and computational load while still achieving superior performance.
>
> >*"The novelty of the paper is limited as it combines existing methods such as "saliency" and "associative memory" for the CL problem."*
>
> **A4.**
> In response to the reviewer's comment on the perceived limited novelty of the paper due to the combination of existing methods - saliency and associative memory - for the Continual Learning (CL) problem, we would like to highlight the unique contributions of our work:
>
> 1. **Innovative Integration for CL**: While it's true that both saliency methods and associative memory have been individually explored in the image retrieval literature, their integration in the context of Continual Learning (CL) represents a significant advancement. Continual Learning, with its unique challenges such as catastrophic forgetting and memory efficiency, requires a nuanced approach. Our work is not just a simple combination of these two methods; it's an innovative adaptation to address the specific needs and challenges of CL.
>
> 2. **Context-Specific Adaptation**: The application of these mechanisms to CL is not trivial. It required significant adjustments and innovations to make them suitable for the dynamic and evolving nature of CL. This adaptation includes the development of a novel method for efficient and selective memory encoding and retrieval, which is critical for managing the complex data streams inherent in CL.
>
> 3. **Unique Problem-Solving Approach**: The problem set in CL is fundamentally different from traditional image retrieval. In CL, the model continuously learns from a stream of data while retaining previously acquired knowledge. This poses unique challenges, such as avoiding catastrophic forgetting, which are not present in conventional image retrieval scenarios. Our method addresses these challenges by leveraging saliency and associative memory in a novel way, specifically tailored for CL.
>
> 4. **Enhanced Performance and Efficiency**: The integration of saliency-guided memory encoding and associative memory retrieval in our SHARC framework enhances memory efficiency and reduces the computational overhead, which is a significant contribution to the field of CL. This efficiency is crucial in CL where models need to be highly adaptive yet resource-efficient.
>
> 5. **Extending Beyond Existing Literature**: While prior works have explored these mechanisms separately, our work synergizes them in a novel framework specifically for CL. This not only advances the state-of-the-art in CL but also opens up new avenues for future research in combining these techniques for other complex learning scenarios.
>
> In summary, while our work builds upon existing methods, its novelty lies in the unique integration and adaptation of these methods to address the specific challenges of Continual Learning. This is not merely an application of existing techniques to a new problem but a thoughtful and innovative reimagining of these techniques to create a solution that is greater than the sum of its parts.

---

### Official Review · Reviewer_prAb · 2023-10-29

**Soundness:** 2 fair
**Presentation:** 3 good
**Contribution:** 2 fair
**Rating:** 3
**Confidence:** 5

**Summary:**

The paper introduces a novel framework called Saliency-Guided Hidden Associative Replay for Continual Learning (SHARC) to address the challenge of catastrophic forgetting in continual learning.

In this paper, we provide a plugin to enhance the performance of replay-based continual learning methods to improve storage efficiency.

The paper proposes the SHARC framework, which combines associative memory with replay-based strategies. SHARC encodes and archives salient data segments using sparse memory encoding. By leveraging associative memory paradigms, SHARC introduces a content-focused memory retrieval mechanism, promising quick and accurate recall.

 The associative memory creates an additional memory footprint and consumes a lot of computing resources for updating it.

The paper presents extensive experimental results that demonstrate the effectiveness of SHARC for various continual learning tasks.

**Strengths:**

The proposed method can be seamlessly adapted to any replay-based approach, improving their performance in various continual learning scenarios.

The experimental results provide evidence of the effectiveness of SHARC in improving the performance of replay-based methods.

**Weaknesses:**

Lack of detailed network and hyper-parameter configuration, especially, for associative memory networks.

The lack of recent baselines in the experiment, most of the baselines used were proposed two or three years ago.

The associative memory A(x,ω) is implemented as a recurrent or feed-forward neural network. The associative memory creates an additional memory footprint, and consumes a lot of computing resources for updating it.

The experiments in this paper do not seem to be reasonable. The method in this paper introduces an additional associative memory network to store more information for replay, which definitely makes the original method perform better.

**Questions:**

How much memory and computing resources does associative memory take up? Wouldn't it be better if those extra storage resources were used to store more exemplars? Please make a comparison.

---

> ### Author Response · Authors · 2023-11-23
> **Detailed Experiment Settings and Model Capacity Explanation(Part1)**
>
> >*“Lack of detailed network and hyper-parameter configuration, especially, for associative memory networks.“*
>
> **A1.**
> By expanding on the previous list of hyperparameters, we aim to offer a more comprehensive illustration of the parameter settings for our associative memory network. This additional information will enable a deeper analysis and evaluation of the different methods employed. We believe that a thorough parameterization is essential for replicability, comparability, and overall confidence in the results obtained. With this in mind, we present the following comprehensive parameterization for your consideration.
>
> For the different methods, we use the same parameter settings for our associative memory network:
>
> Bpcn_args:
> #### Model Configs
>
> | Parameter         | Value      |
> |-------------------|------------|
> | memory_strength   | 1          |
> | n-models          | 1          |
> | n-layers          | 2          |
> | h-dim             | 256        |
> | sigma-prior       | 1.0        |
> | sigma-obs         | 0.01       |
> | beta-forget       | 0.1        |
> | beta-noise        | 0.1        |
> | scale-layer       | store_true |
> | bias              | store_true |
> | n-elbo-particles  | 1          |
>
> #### Training Configs
>
> | Parameter             | Value      |
> |-----------------------|------------|
> | weight-lr             | 0.0001     |
> | activation-lr         | 0.01       |
> | activation-optim      | adam       |
> | T-infer               | 500        |
> | n-proposal-samples    | 1          |
> | n-repeat              | 1          |
> | resample              | store_true |
>
> Methods_args:
>
> | Method   | Theta | Learning Rate | Minibatch Size | Batch Size | Number of Epochs | Samples per Task | Number of Tasks |
> |----------|-------|---------------|----------------|------------|------------------|------------------|-----------------|
> | er_bpcn  | 200   | 0.1           | 10             | 10         | 1                | 10000            | 5               |
> | er_bpcn  | 500   | 0.1           | 10             | 10         | 1                | 10000            | 5               |
> | er_bpcn  | 1000  | 0.1           | 10             | 10         | 1                | 10000            | 5               |
>
>
>
> | Method   | Theta | Learning Rate | Number of Epochs | Beta | Gamma | Minibatch Size | Batch Size | Batch Number | Samples per Task | Number of Tasks |
> |----------|-------|---------------|------------------|------|-------|----------------|-------------|--------------|------------------|-----------------|
> | mer_bpcn | 200   | 0.1           | 1                | 1    | 1     | 10             | 10          | 1            | 10000            | 5               |
> | mer_bpcn | 500   | 0.1           | 1                | 1    | 1     | 10             | 10          | 1            | 10000            | 5               |
> | mer_bpcn | 1000  | 0.1           | 1                | 1    | 1     | 10             | 10          | 1            | 10000            | 5               |
>
>
>
> | Method   | Theta | Learning Rate | Gamma | Minibatch Size | Batch Size | Number of Epochs | Samples per Task | Number of Tasks |
> |----------|-------|---------------|-------|----------------|------------|------------------|------------------|-----------------|
> | gem_bpcn | 200   | 0.1           | 0.5   | 10             | 10         | 1                | 10000            | 5               |
> | gem_bpcn | 500   | 0.1           | 0.5   | 10             | 10         | 1                | 10000            | 5               |
> | gem_bpcn | 1000  | 0.1           | 0.5   | 10             | 10         | 1                | 10000            | 5               |
>
>
>
> | Method    | Theta | Learning Rate | Minibatch Size | Batch Size | Number of Epochs | Samples per Task | Number of Tasks |
> |-----------|-------|---------------|----------------|------------|------------------|------------------|-----------------|
> | agem_bpcn | 200   | 0.1           | 10             | 10         | 1                | 10000            | 5               |
> | agem_bpcn | 500   | 0.1           | 10             | 10         | 1                | 10000            | 5               |
> | agem_bpcn | 1000  | 0.1           | 10             | 10         | 1                | 10000            | 5               |
>
>
>
> | Method     | Theta | Alpha | Beta | Learning Rate | Minibatch Size | Batch Size | Number of Epochs | Samples per Task | Number of Tasks |
> |------------|-------|-------|------|---------------|----------------|------------|------------------|------------------|-----------------|
> | derpp_bpcn | 200   | 0.1   | 0.5  | 0.1           | 10             | 10         | 1                | 10000            | 5               |
> | derpp_bpcn | 500   | 0.1   | 0.5  | 0.1           | 10             | 10         | 1                | 10000            | 5               |
> | derpp_bpcn | 1000  | 0.1   | 0.5  | 0.1           | 10             | 10         | 1                | 10000            | 5               |

---

> ### Author Response · Authors · 2023-11-23
> **Detailed Experiment Settings and Model Capacity Explanation(Part2)**
>
> | Method     | Theta | Reg Weight | Stable Model Update Freq | Stable Model Alpha | Plastic Model Update Freq | Plastic Model Alpha | Learning Rate | Minibatch Size | Batch Size | Number of Epochs | Samples per Task | Number of Tasks |
> |------------|-------|------------|-------------------------|--------------------|--------------------------|---------------------|----------------|----------------|------------|------------------|------------------|-----------------|
> | clser_bpcn | 200   | 0.15       | 0.1                     | 0.999              | 0.3                      | 0.999               | 0.1            | 10             | 10         | 1                | 10000            | 5               |
> | clser_bpcn | 500   | 0.15       | 0.1                     | 0.999              | 0.3                      | 0.999               | 0.1            | 10             | 10         | 1                | 10000            | 5               |
> | clser_bpcn | 1000  | 0.15       | 0.1                     | 0.999              | 0.3                      | 0.999               | 0.1            | 10             | 10         | 1                | 10000            | 5               |
>
>
>
>
> >*"The experiments in this paper do not seem to be reasonable. The method in this paper introduces an additional associative memory network to store more information for replay, which definitely makes the original method perform better."*
>
> **A2.**
> The role of our associative memory network is to reduce the memory consumption of whole images by converting them into individual feature maps and depositing them into episodic memory after going through our GradCAM method. This reduction in memory usage is significant.
>
> However, it is important to note that even with the introduction of an associative memory network, the amount of information we introduce is much less compared to methods such as ER and MER, which store images directly into episodic memory.
>
> The reduction in information is due to the transformation of the original image into a feature map representation. Unlike the complete image, which includes all the pixel-level details and color information, a feature map represents a condensed version that highlights specific visual features and patterns. This reduction in information allows for more efficient storage and processing, making it an effective strategy for reducing memory consumption in our associative memory network.
>
> In addition, our associate memory, or BayesPCN, is a compact network architecture comprising only a few layers of Multi-Layer Perceptron (MLP). While BayesPCN plays a crucial role, it is important to note that the memory it provides is relatively limited when compared to the vast amount of information present in the complete image dataset.
>
> >*"How much memory and computing resources does associative memory take up? Wouldn't it be better if those extra storage resources were used to store more exemplars? Please make a comparison."*
>
> **A3.**
> Take the method ER with SHARC as an example. In the specific case of theta=0.2, where only 20% of the original image is retained, if we were to deposit all the complete images inside the episodic memory, it would require approximately 7.5 million parameters. However, with our SHARC method, the episodic memory is represented by a single feature map, resulting in a reduction to only about 1.3 million parameters. This optimization effectively drops about 6 million parameters, and the computation involved is only 17% of the original.
>
> Comparatively, the ResNet18 architecture, which is commonly used as a backbone in deep learning models, consists of around 11 million parameters. In contrast, the associative memory network that we introduced contains only about 4 million parameters. By adopting our approach, we achieve a memory consumption reduction of approximately 11% compared to the original tasker, while also obtaining higher accuracy.
>
> These results clearly demonstrate the efficiency and accuracy of our method. By optimizing the episodic memory representation and introducing the associative memory network, we significantly reduce the parameter count and computational load while still achieving superior performance.

---

### Official Review · Reviewer_WbsA · 2023-10-31

**Soundness:** 1 poor
**Presentation:** 3 good
**Contribution:** 3 good
**Rating:** 3
**Confidence:** 5

**Summary:**

In conventional replay-based continual learning methods, raw/ entire data are stored and recalled during replay which is biologically implausible and memory intensive. This work presents a biologically plausible framework where partial salient data are stored and complete data are retrieved during replay. It utilizes sparse memory encoding to store partial information and a content-based memory retrieval mechanism to recover complete information. It enables efficient memory storage and archives better recall accuracy than generative models. This work has potential to perform effectively in highly memory-constrained applications.

**Strengths:**

This paper presents bio-inspired perspectives to store hidden sparse representations and associate memory based recall. This resembles how humans and animals learn by compressing information.

Innovative approach of storing and retrieving rehearsal data for memory efficient replay and mitigating catastrophic forgetting.

Saliency based approach to store sparse information which leads to increased memory efficiency.

Associate memory based retrieval offers fast and efficient recall and higher noise tolerance. This is an innovative approach to reduce memory footprint and computational overhead in continual learning.

Memory-forgetting mechanism to remove more data from old tasks than new tasks.

Several SOTA methods show improved performance when combined with proposed method, SHARC

**Weaknesses:**

Lack of experiments on high dimensional and large scale datasets e.g., ImageNet-1K. Many algorithms do not scale for large numbers of classes and high-dimensional inputs. It is 2023 and people have been using ImageNet for continual learning for at least 7 years. I'm assuming this is because they are starting with an ImageNet pre-trained backbone, but that is also a problem given the datasets studied. Mini-imageNet is not an appropriate test set using an ImageNet pre-trained backbone. MNIST and CIFAR are also extremely inappropriate. This is throwing a comparatively very powerful network at toy problems, where training its output layer alone likely yields extremely high results.

The model is only tested for an extreme edge case in continual learning (class incremental learning). Other distributions need to be studied, IID, etc. Given the neuroinspiration, this is especially important, but it conflates the goals of continual learning (knowledge accumulation over time) with the test (learning classes one at a time). An ideal continual learner should be robust to any data orderings including class incremental learning and IID.

Limited representation learning. It keeps the feature backbone frozen and trains the classifier head and associative memory network. Thus the model has limitations in learning representations in hidden layers which might be necessary for learning new tasks.
It is unclear how model depth impacts retrieval performance, for example when we want to store and retrieve information in the earlier layers close to input.

Given ImageNet-1K pretrained backbone, selected datasets e.g., MNIST, CIFAR-10 / 100, and mini-ImageNet seem less challenging for a continual learner. It is also unclear how ImageNet-1K (224x224) pretrained network is used for small datasets consisting of lower resolution images (32x32).

Since SHARC requires training associate memory unlike comparison methods. Comparing methods based on the same bounded compute (same amount of training updates) will be fairer.

Sometimes SHARC under-performs some baselines (Table 2 and Table 4). It is unclear if SHARC provides consistent performance gain across CL settings / methods/ datasets/ buffer sizes. It is claimed that DER++ equipped with SHARC achieves a 45.5% improvement in accuracy on S-CIFAR-100 but results in Table 2 do not support this claim.

The experiments and evaluation leave a great deal to be desired. Using an ImageNet-1K pre-trained backbone is fine, but then the experiments would need to be appropriate, for example, learning a dataset like iNaturalist or Places-365, and then trying multiple different distributions, including incremental class learning. Also, more experimental comparisons against recent methods and benchmarking in a fair way where all models are compared with the same setup would be more sound.

The method needs to demonstrate some sort of efficiency or other value in some sense.

The method is interesting, but the evaluation and experimental confounds mean the paper is not ready for publication. I encourage the authors to redesign their experiments, eliminate confounds, study multiple distributions, and to study much larger and more appropriate datasets. It requires relatively few resources to train an ImageNet-1K model and can be done with cloud computing for very little money.

**Questions:**

In Fig.1, spatial information was retrieved but in the main experiment channel information was recovered. If you mask out spatial information, can you apply similar associate memory to retrieve complete information?

What happens if you train more layers of DNN besides the final layer?

What is the computational overhead for training associate memory? Does associate memory increase inference cost?

How do you initialize the last layer before continual learning begins?

How do you use ImageNet-1K (224x224) pretrained network for small datasets consisting of lower resolution images (32x32)?

Besides Fig.1, do you have results to support the claims about fast and efficient recall and noise tolerance?

---

> ### Comment · Reviewer_WbsA · 2023-11-22
>
> The authors have not provided a rebuttal, but I did review the comments from the other reviewers. My rating is unchanged.

---

### Official Review · Reviewer_3N7Y · 2023-11-03

**Soundness:** 3 good
**Presentation:** 3 good
**Contribution:** 3 good
**Rating:** 6
**Confidence:** 2

**Summary:**

The paper introduces the Saliency-Guided Hidden Associative Replay (SHARC) framework, which combines associative memory with replay-based strategies to address catastrophic forgetting in Continual Learning. Firstly, SHARC archives only the most salient segments of data through sparse memory encoding, making it memory-efficient. Secondly, this paper proposes a content-centric memory retrieval module inspired by associative memory, enabling swift and impeccable recall capabilities. Extensive experimental results demonstrate the efficacy and superiority of the proposed SHARC framework for various continual learning tasks .

**Strengths:**

The paper introduces the novel Saliency-Guided Hidden Associative Replay (SHARC) framework, which combines associative memory with replay-based strategies to address catastrophic forgetting in Continual Learning.
The proposed SHARC framework demonstrates its effectiveness through extensive experimental results on various continual learning tasks, showcasing its superiority in mitigating forgetting and achieving better recall.
The structure of SHARC is sparsity, which is hardware-friendly and can lead to memory cost reduction instantly.

**Weaknesses:**

The paper does not provide a comprehensive comparison with existing replay-based methods for Continual Learning, making it difficult to assess the superiority of the proposed framework.
While the paper introduces a content-focused memory retrieval mechanism, it lacks detailed explanation and analysis of how this mechanism works and its impact on recall performance.

**Questions:**

It would be beneficial to include a comprehensive comparison with existing replay-based methods for Continual Learning to highlight the advantages and limitations of the proposed framework.
Could the authors provide a more detailed explanation and analysis of the content-focused memory retrieval mechanism introduced in the SHARC framework? This would help in understanding how this mechanism works and its impact on recall performance.

---

> ### Author Response · Authors · 2023-11-23
> **Additional Experiment and Methodology Explanation**
>
> >**Q1.** *The paper does not provide a comprehensive comparison with existing replay-based methods for Continual Learning, making it difficult to assess the superiority of the proposed framework.*
>
> **A1.**
> Thank you for your suggestion. Additional results have been added to the general response **above**.
>
> >**Q2.** *While the paper introduces a content-focused memory retrieval mechanism, it lacks detailed explanation and analysis of how this mechanism works and its impact on recall performance.*
>
> **A2.**
> Associative memory is based on attractor dynamics, which refers to neuronal network dynamics dominated by groups of persistently active neurons. Generally, such persistent activation is associated with an attractor state of the dynamics, often in the form of a fixed point. This type of network can be used to implement associative memory by allowing the network's attractors to correspond to the vectors we want to store. This approach supports memory for specific items and differs from semantic memory in that it stores items quickly but does not represent the semantic structure of the data. Instead, attractor dynamics resembles working and episodic memory. Like episodic memory, it acts as an associative memory, returning stored values when triggered with the right cues.
> The Hopfield network, originally proposed in 1982 (Hopfield, 1982), is a recurrent neural network that implements associative memory using fixed points as attractors. The function of the associative memory is to recognize previously learned input vectors, even in cases where some noise has been added. To achieve this function, every neuron in the network is connected to all the others (see Fig. 2.4(a)). Each neuron outputs discrete values, normally 1 or -1, according to the following equation:
> x_i(t+1) = sign(Σ_{j=1}^{N} w_{ij}x_j(t))
> where x_i(t) is the state of the i-th neuron at time t, and N is the number of neurons. The Hopfield network has a scalar value associated with the state of all neurons x, referred to as the "energy" or Lyapunov function:
> E(x) = -1/2 Σ_{i=1}^{N} Σ_{j=1}^{N} w_{ij}x_i x_j
> If we want to store Q patterns x_p, p=1,2,...,Q, we can use the Hebbian learning rule (Hebb, 1962) to assign the values of the weights as follows:
> w_{ij} = Σ_{p=1}^{Q} x_p^i x_p^j
> This is equivalent to setting the weights to the elements of the correlation matrix of the patterns. Upon presentation of an input to the network, the activity of the neurons can be updated (asynchronously) according to Eq. (2.22) until the energy function has been minimized (Hopfield, 1982). Hence, repeated updates would eventually lead to convergence to one of the stored patterns. However, the network may possibly converge to spurious patterns (different from the stored patterns) as the energy in these spurious patterns is also a local minimum.

---

> ### Comment · Reviewer_3N7Y · 2023-12-03
> **Keep my score**
>
> I appreciate the author's thorough response to my questions. After revisiting the paper and the rebuttal, I have decided to maintain my current rating.

---

### Author Response · Authors · 2023-11-23
**Global Response from Author (Additional Results)**

Thank you for your time and dedication to this review process. We are grateful for the opportunity to benefit from your expertise, and we believe your contributions will enhance the overall quality of the SHARC.

To address the lack of state-of-the-art methods and improperly pre-trained weights, we re-ran some of the experiments on the split CIFAR100 and split Mini-ImageNet datasets. The initial weights are pre-trained on 50 categories that do not overlap with those in Mini-ImageNet. The results are shown in the following table, where CLSER[1] and OCDNet[2] were proposed in 2022. The buffer size is 200 samples and the metric we use is accuracy. Models in bold are equipped with SHARC.

| Model    | S-CIFAR-100 (Class-IL)  | S-CIFAR-100 (Task-IL) |  S-Mini-ImgNet (Class-IL)  | S-Mini-ImgNet (Task-IL) |
| :-----------: | :-------------: | :----------: | :----------: | :---------------: |
| GEM         | 14.51 ± 2.53  |  77.61 ± 1.57            |   17.25 ± 2.31   |    62.57 ± 1.72  |
| **GEM+** | 16.32 ± 2.84  |  78.72 ± 1.72            |   18.01 ± 1.46    |   64.32 ± 0.93   |
| ER         |   13.31 ± 2.73   |  78.31 ± 0.72    |    16.28 ± 2.52   |   63.34 ± 1.12  |
| **ER+** |   16.38 ± 3.61  |   79.23 ± 0.88  |    17.32 ± 2.12     |   65.76 ± 0.62  |
| DERPP         |  11.23 ± 3.13|   78.12 ± 1.43  |  17.93 ± 1.71   |  64.63 ± 2.35  |
| **DERPP+** |  17.01 ± 2.91 |  78.99 ± 1.01  |  18.62 ± 2.72  |  65.12 ± 1.58  |
| CLSER         |  14.46 ± 1.73 |  77.47 ± 0.37  |  17.30 ± 1.34  |  64.51 ± 1.33  |
| **CLSER+** |  15.36 ± 1.57 |   78.34 ± 0.65  |   17.95 ± 0.37 |  65.32 ± 0.61 |
| OCDNet         |   15.36 ± 1.82  |  77.13 ± 2.68 |  17.83 ± 1.77  |  64.81 ± 1.27  |
| **OCDNet+** |   17.45 ± 1.58  |   77.41 ± 1.67    | 18.71 ± 1.48      |  65.83 ± 0.66   |

Overall, the results still prove the effectiveness of SHARC in preventing forgetting.

[1] Learning Fast, Learning Slow: A General Continual Learning Method based on Complementary Learning System. ICLR 2022

[2] Learning from Students: Online Contrastive Distillation Network for General Continual Learning. IJCAI 2022

---

### Meta-Review · Area_Chair_otv5 · 2023-12-06

**Metareview:**

The paper presents a saliency-guided hidden associative replay method for continual learning.
4 out of the 5 reviewers gave scores below 5; while one reviewer gave a score of 6 but with low confidence. The responses from the authors did not fully address all the reviewers' concerns. Given the current status of the paper (see main weaknesses below), AC recommends rejections.

Strengths:
1. The paper is well-written and easy to understand.
2. The idea about interactions among saliency, continual learning, and associative memory is interesting.

Weaknesses:
1. lack of comprehensive comparisons with different types of continual learning baselines, such as recent replay methods, architecture expansion methods, regularization methods, and prompt-based methods for pre-trained networks.
2. The saliency is obtained by gradient backdrop. Thus, there is a lack of alignment with human saliency maps on these images. The claim about human memory inspiration seems to be hand-wavy.
3. the expeirments are too simplified. Benchmarking on larger complex datasets over longer tasks, such as imagenet 1k, is missing. There is also no benchmarking in other problem settings in continual learning, such as task incremental and data incremental.
4. the model uses pre-trained feature extractors from imagenet. This defeats the purpose of continual learning as it is tricky to justify whether the classes to be learned have already been seen during pre-training. As the reviewer recommended, cross-domain continual learning would be a better venue to test the effectiveness of the proposed approach.

**Justification For Why Not Higher Score:**

see weakness above

**Justification For Why Not Lower Score:**

NA

---

### Decision · Program_Chairs · 2024-01-16

Reject